# USP44 regulates irradiation-induced DNA double-strand break repair and suppresses tumorigenesis in nasopharyngeal carcinoma

Yang Chen[1,3], Yin Zhao [1,3], Xiaojing Yang[1,3], Xianyue Ren[2,3], Shengyan Huang[1], Sha Gong[1], Xirong Tan[1], Junyan Li[1], Shiwei He[1], Yingqin Li[1], Xiaohong Hong[1], Qian Li[1], Cong Ding[1], Xueliang Fang[1], Jun Ma [1] & Na Liu [1✉]

Radiotherapy is the primary treatment for patients with nasopharyngeal carcinoma (NPC), and approximately 20% of patients experience treatment failure due to tumour radio-resistance. However, the exact regulatory mechanism remains poorly understood. Here, we show that the deubiquitinase *USP44* is hypermethylated in NPC, which results in its down-regulation. USP44 enhances the sensitivity of NPC cells to radiotherapy in vitro and in vivo. USP44 recruits and stabilizes the E3 ubiquitin ligase TRIM25 by removing its K48-linked polyubiquitin chains at Lys439, which further facilitates the degradation of Ku80 and inhibits its recruitment to DNA double-strand breaks (DSBs), thus enhancing DNA damage and inhibiting DNA repair via non-homologous end joining (NHEJ). Knockout of TRIM25 reverses the radiotherapy sensitization effect of USP44. Clinically, low expression of USP44 indicates a poor prognosis and facilitates tumour relapse in NPC patients. This study suggests the USP44-TRIM25-Ku80 axis provides potential therapeutic targets for NPC patients.

[1] State Key Laboratory of Oncology in South China; Collaborative Innovation Center of Cancer Medicine; Guangdong Key Laboratory of Nasopharyngeal Carcinoma Diagnosis and Therapy, Sun Yat-sen University Cancer Center, Guangzhou 510060, P. R. China. [2] Guangdong Provincial Key Laboratory of Stomatology, Guanghua School of Stomatology, Hospital of Stomatology, Sun Yat-sen University, Guangzhou 510055, P. R. China. [3] These authors contributed equally: Yang Chen, Yin Zhao, Xiaojing Yang, Xianyue Ren. ✉email: liun1@sysucc.org.cn

Nasopharyngeal carcinoma (NPC) is an epithelial tumour arising from the nasopharyngeal mucosa with a unique geographical distribution. It is prevalent in South China, South-Eastern Asia and North Africa[1,2]. Radiotherapy is the primary therapeutic method for NPC because the disease is highly sensitive to ionising radiation[2]. Recently, the developments of more accurate tumour localisation methods, better radiotherapy techniques and combined therapy have greatly improved patient survival. However, ~20% of patients suffer regional recurrence or distant metastasis due to radioresistance[3–5]. However, the exact regulatory mechanisms underlying radioresistance in NPC are poorly understood.

DNA double-strand breaks (DSBs) are the most critical type of DNA damage induced by irradiation (IR), and the majority of DSBs are repaired via the non-homologous end joining (NHEJ) pathway[6,7]. The Ku80-Ku70 heterodimer binds rapidly and tightly to the ends of DSBs and further recruits many other factors required for NHEJ-mediated DNA repair, including DNA-dependent protein kinase catalytic subunit (DNA-PKcs), the XRCC4–LIG4–XLF ligation complex, and APTX and APTF proteins; thus, Ku80 plays an essential role in the initiation of the NHEJ-mediated DNA repair pathway[8,9]. Ku80 is tightly regulated by ubiquitination and deubiquitination mediated by E3 ubiquitin ligases (E3s) and deubiquitinating enzymes (DUBs), respectively. For example, the ubiquitination of Ku80 is mediated by multiple E3s, including RING finger domain-containing protein (RNF) RNF8[10], RNF126[11] and RNF138[12]. Conversely, the ubiquitin carboxyl-terminal hydrolase L3 (UCHL3), which belongs to the DUB family, can directly deubiquitylate Ku80[13]. However, how Ku80 is recruited to damaged DNA remains obscure.

DNA methylation, a type of epigenetic modification, is closely associated with tumour initiation and progression, especially in NPC[14–16]. Frequent methylation of the CpG islands of the ubiquitin-specific protease (USP) USP44 is an early event in colorectal neoplasia[17]. However, the functions and mechanisms of USP44 in NPC have not yet been investigated. USP44 is involved in cell cycle regulation, cell differentiation and DNA repair processes[18,19]. For example, USP44 acts as a tumour suppressor by inhibiting the activation of APC to prevent the missegregation of chromosomes[20,21]. USP44 can also regulate stem cell differentiation by reversing the mono-ubiquitination of H2B-K120[22]. In addition, in the DSB response, USP44 counteracts the RNF168-mediated polyubiquitination of histone H2A to inhibit the recruitment of downstream repair factors[23].

Here, we show that hypermethylation of USP44 promotes radiotherapy resistance in NPC. USP44 is hypermethylated in NPC, which is associated with its downregulation. USP44 enhances the sensitivity of NPC cells to radiotherapy in vitro and in vivo through the USP44-TRIM25-Ku80 axis. USP44 recruits and stabilises the tripartite motif-containing (TRIM) protein TRIM25 by removing its K48-linked polyubiquitin chains at Lys439, which further facilitates the degradation of Ku80 and inhibits its recruitment to DSBs, thus enhancing DNA damage and inhibiting NHEJ-mediated DNA repair. Low expression of USP44 is associated with tumour relapse and a poor prognosis in NPC patients. The USP44-TRIM25-Ku80 axis provides potential targets for NPC treatment and prognostic prediction.

## Results

### Promoter hypermethylation of USP44 downregulates its expression in NPC.
Our previous methylation microarray study (GSE52068) analysed genome-wide DNA methylation between normal nasopharyngeal ($n = 24$) and NPC tumour ($n = 24$) samples[24], from which we identified seven hypermethylated CpG sites in the promoter of USP44 (Fig. 1a). Among the 7 CpG sites,

site cg00927554 was the most hypermethylated (Supplementary Fig. 1a), and this result was confirmed in another published microarray dataset (GSE62336, Supplementary Fig. 1a) from Hong Kong. Thus, we selected it for further validation by bisulfite pyrosequencing (Fig. 1b). The cg00927554 site of the USP44 promoter was more significantly hypermethylated in NPC tissues than in normal tissues (Fig. 1c, d). The average methylation rate of this site was more than 90% in NPC cell lines but was only ~10% in normal NP69 cells (Fig. 1e). In addition, we found that NPC cell lines and tissue samples had much lower USP44 mRNA and protein expression levels than the immortalised nasopharyngeal epithelial NP69 cells and normal tissue samples (Fig. 1f–i). The demethylating drug DAC was used to verify whether the downregulation of USP44 resulted from the hypermethylation of its promoter. DAC treatment substantially decreased USP44 methylation levels but increased USP44 mRNA levels in NPC cells compared with NP69 cells (Fig. 1j, k). Moreover, TCGA database analysis using the GEPIA tool showed USP44 promoter hypermethylation and downregulated mRNA expression, and this negative correlation was observed in eight other solid tumour types (Supplementary Fig. 1b–d). Taken together, these data illustrate that the promoter hypermethylation of USP44 results in its downregulation in NPC.

### USP44 enhances the radiosensitivity of NPC cells in vitro.
Through Gene Set Enrichment Analysis (GSEA) of the GSE12452 dataset, we identified that compared to NPC samples with high USP44 expression, those with low USP44 expression were remarkably enriched in gene sets related to radiation response pathways (Fig. 2a). To further investigate the effect of USP44 after DNA damage in NPC cells, we constructed SUNE1 and HONE1 cells with stable overexpression or transient knockdown of USP44 (Supplementary Fig. 2a, b). Overexpression of USP44 severely impeded the colony formation and cell proliferation of NPC cells after IR (Fig. 2b and Supplementary Fig. 2c). Conversely, knockdown of USP44 in NPC cells improved cell survival and proliferation after DNA damage caused by IR (Supplementary Fig. 2d, e), which were confirmed by knockout of USP44 expression in SUNE1 cells (Supplementary Fig. 2f).

One of the most common effects of IR is cell cycle arrest[25]. An increasing proportion of cells in the G2/M phase indicates that cells are more sensitive to IR[26–28]. DNA damage after IR also leads to a strong cell apoptosis response[29]. We found that the combination of IR and USP44 overexpression significantly induced the G2/M phase arrest and apoptosis of NPC cells (Fig. 2c, d). However, knockdown of USP44 in NPC cells upon IR significantly reduced G2/M phase arrest and apoptosis (Supplementary Fig. 3a, b). The microtubule poison nocodazole can arrest cells in the G2/M phase when H3S10$_P$ is highly abundant[30–34]. We arrested cells in the G2/M phase with nocodazole and investigated the effect of IR treatment and USP44 knockout on G2/M cell cycle arrest. Our results revealed that the percentage of H3S10$_P$ positive cells was obviously enhanced upon IR induction, which could lead to DNA damage and arrest cells in the G2/M phase. Knockout of USP44 decreased the percentage of H3S10$_P$ positive cells and inhibited the IR-induced G2/M cell cycle arrest (Supplementary Fig. 3c). These results elucidated that USP44 sensitises NPC cells to IR through G2/M phase arrest and apoptosis induction, indicating an essential role of USP44 in the DNA damage response.

### USP44 promotes the degradation of Ku80 by enhancing its ubiquitination.
Mass spectrometry analysis identified Ku80, which had the greatest number of peptide ions matched to USP44, as the potential target of USP44 (Fig. 3a, Supplementary

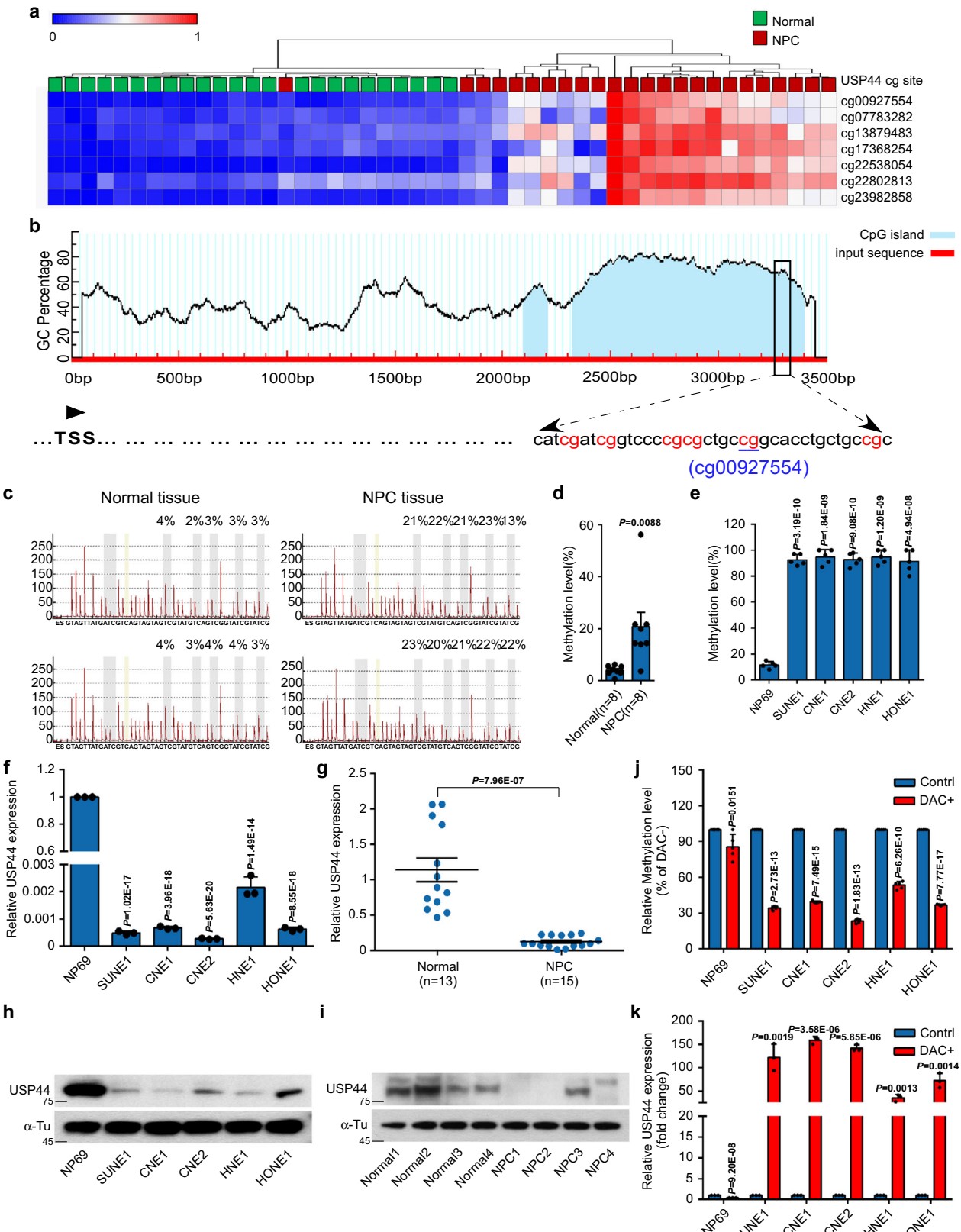

Fig. 4a and Supplementary Table 1). Co-IP also verified the exogenous interaction between USP44 and Ku80 (Fig. 3b). Immunofluorescence staining further confirmed the co-localisation of HA-USP44 and Ku80 in the nucleus (Fig. 3c). Ku80 and Ku70, as a complex, recognise DSBs and recruit other NHEJ proteins in the DNA repair process[35]. Therefore, we next

sought to determine how the interaction between USP44 and Ku80 proteins affects radiosensitization in NPC. We found that overexpression of USP44 decreased the protein expression level of Ku80 in a dose-dependent manner but did not affect its mRNA expression level (Fig. 3d and Supplementary Fig. 4b). Over-expression of USP44 significantly promoted the degradation of

**Fig. 1 Promoter hypermethylation of *USP44* downregulates its expression in NPC. a** Heatmap clustering of seven hypermethylated CpG sites in the CpG islands of *USP44* in normal nasopharyngeal epithelial tissues (*n* = 24) and NPC tissues (*n* = 24). Columns: individual samples; rows: CpG sites; blue: low methylation; red: high methylation. **b** Schematic illustration of the bisulfite pyrosequencing region in the *USP44* promoter. Red region: input sequence; blue region: CpG islands; TSS: transcription start site; red text: CG sites used for bisulfite pyrosequencing; blue text: the most significantly altered CG site in the *USP44* promoter. **c, d** Bisulfite pyrosequencing analysis of the *USP44* promoter region (**c**) and statistical analysis of methylation levels (**d**) in normal (*n* = 8) and NPC (*n* = 8) tissues. **e** The methylation levels of the *USP44* promoter region between NP69 and NPC cell lines (SUNE1, CNE1, CNE2, HNE1, and HONE1) were determined through bisulfite pyrosequencing analysis. **f, g** RT-PCR analysis of relative USP44 mRNA expression in the NP69 cell line and NPC cell lines (**f**) and in normal (*n* = 13) and NPC (*n* = 15) tissues (**g**). **h, i** Representative western blot analysis of USP44 protein expression in NP69 cells and NPC cell lines (**h**), together with normal and NPC tissues (**i**). **j, k** *USP44* methylation levels measured by bisulfite pyrosequencing analysis (**j**) and relative USP44 mRNA levels measured by RT-PCR analysis (**k**) in NP69 cells and NPC cell lines with (DAC+) or without (DAC−) DAC treatment. Data in **d** and **g** are presented as the mean ± SEM, and those in **e**, **f**, **j**, and **k** are presented as the mean ± SD; the *P* values were determined using the two-tailed Student's *t*-test; *n* = 3 independent experiments. Source data are provided as a Source Data file.

Ku80, and knockout of USP44 inhibited the degradation of endogenous and exogenous Ku80 through treatment with CHX, which means that USP44 could shorten the half-life of the Ku80 protein (Fig. 3e and Supplementary Fig. 4c, d). To further investigate whether USP44 promotes the degradation of Ku80 through the ubiquitin-proteasome pathway or lysosomal pathway, we treated HEK293T cells with MG132, a proteasome inhibitor, and CQ, a lysosome inhibitor, after co-transfection with USP44 and Ku80, and we found that USP44-mediated destabilization of Ku80 was reversed by MG132 but not by CQ, suggesting that USP44 downregulates the Ku80 protein through the ubiquitin-proteasome pathway (Fig. 3f). USP44 belongs to the DUB family and possesses ubiquitin hydrolase activity[22,23]. We, therefore, examined the effects of USP44 on the ubiquitination of Ku80 and found that overexpression of USP44 surprisingly increased the polyubiquitination of Ku80 (Fig. 3g and Supplementary Fig. 4e). These results suggest that USP44 promotes the polyubiquitination of Ku80, which results in its degradation via the ubiquitin-proteasome pathway.

**USP44 recruits TRIM25 to ubiquitinate Ku80 and further leads to its degradation**. The above results showed that USP44 promotes the ubiquitination and degradation of Ku80. However, USP44 acts as a DUB and usually stabilises the target protein through the deubiquitination process[36]. We hypothesised that USP44 may recruit an E3 ligase to promote the ubiquitination and degradation of Ku80. We then found the E3 ligase TRIM25 by mass spectrometry (Fig. 3a, Supplementary Fig. 4a and Supplementary Table 1). Consistent with our hypothesis, TRIM25 could exogenously and endogenously interact with both USP44 and Ku80 (Fig. 3h). The TRIM25 protein contains three domains, including RING finger, protein kinase C-related kinase homology region 1 (HR1) and N-terminal PRY/SPRY domains. Truncation co-IP revealed that USP44 could interact with the HR1 and PRY/SPRY domains of TRIM25 but not the RING finger domain (Supplementary Fig. 5a), suggesting that the HR1 and PRY/SPRY domains are important for the interaction between USP44 and TRIM25. Immunofluorescence staining also revealed co-localisation of USP44, Ku80 and TRIM25 in NPC cells (Supplementary Fig. 5b). Therefore, TRIM25 may function as the E3 ligase between USP44 and Ku80 and eventually lead to the degradation of Ku80.

We then checked whether TRIM25 could affect the stability of Ku80 and found that overexpression of TRIM25 accelerated Ku80 decay rates but did not affect the mRNA expression of Ku80; this effect could be reversed by MG132 but not CQ (Fig. 3i, j and Supplementary Fig. 5c). Overexpression of TRIM25 significantly promoted the degradation of Ku80, and knockout or knockdown of TRIM25 inhibited the degradation of Ku80 through treatment with CHX (Fig. 3k and Supplementary Fig. 5d, e). As expected, overexpression of TRIM25 notably increased the

polyubiquitination of Ku80 (Fig. 3l and Supplementary Fig. 5f). In addition, the knockdown of TRIM25 reversed the increased polyubiquitination of Ku80 caused by ectopic expression of USP44 (Fig. 3m). The above results showed that USP44 recruits TRIM25 to ubiquitinate Ku80 and further leads to its degradation.

**USP44 deubiquitinates and stabilises TRIM25 to promote Ku80 ubiquitination**. TRIM25 recruited by USP44 acts as a scaffold protein between USP44 and Ku80. Next, we wondered whether the stability of TRIM25 is regulated by USP44. To our surprise, overexpression of USP44 stabilised the TRIM25 protein and prolonged its half-life in both HEK293T cells and NPC cells and could be further accumulated in the presence of MG132 (Fig. 4a–c and Supplementary Fig. 6a, b). Furthermore, overexpression of USP44 inhibited the K48-linked but not K63-linked ubiquitination of TRIM25 (Fig. 4d and Supplementary Fig. 6c). Conversely, knockdown or knockout of USP44 enhanced the K48-linked but not K63-linked ubiquitination of TRIM25 (Fig. 4e and Supplementary Fig. 6d). Nevertheless, USP44 (C282A), a deubiquitinase-inactive mutant of USP44, lost its ability to stabilise and deubiquitinate TRIM25 (Fig. 4f, g and Supplementary Fig. 6e), indicating that the ubiquitin hydrolase activity of USP44 is involved in the regulation of TRIM25.

We then generated three K/R substitutions (KR) mutants of TRIM25 according to mass spectrometry analysis (Fig. 4h) for denature-IP assays. When without USP44 overexpression, the ubiquitination levels of all TRIM25 mutants (K283/284R, K439R or K509R) were weaker than that of the wild-type TRIM25 (WT), indicating that all the TRIM25 KR mutants can be ubiquitinated, including K439R. Comparing the amount of K439-ubiquitin smeared in cells with USP44 expressing to the same signal in cells without USP44 expressing clearly demonstrated that USP44 targeted this mutant for deubiquitylation as well. While TRIM25 ubiquitinated on other lysines certainty did not appear resistant to USP44 deubiquitylation, thus USP44 might have the highest activity toward K439 ubiquitination (Fig. 4i). Hence, USP44 recruits TRIM25 and impairs the Lys439-mediated K48-linked ubiquitination of TRIM25 and further inhibits its degradation. We further found that overexpression of USP44 reduced the Ku80 expression with or without IR and knockdown of TRIM25 could rescue the USP44-mediated Ku80 degradation, which was validated by western blot analysis (Fig. 4j and Supplementary Fig. 7a–d). Taken together, our results reveal that USP44 stabilises TRIM25 by removing its K48-linked polyubiquitin chains at K439, which further promotes the ubiquitination and degradation of Ku80.

**Knockout of TRIM25 reverses the radiosensitizing effect of USP44 in vitro**. To validate whether USP44 stabilises TRIM25 to

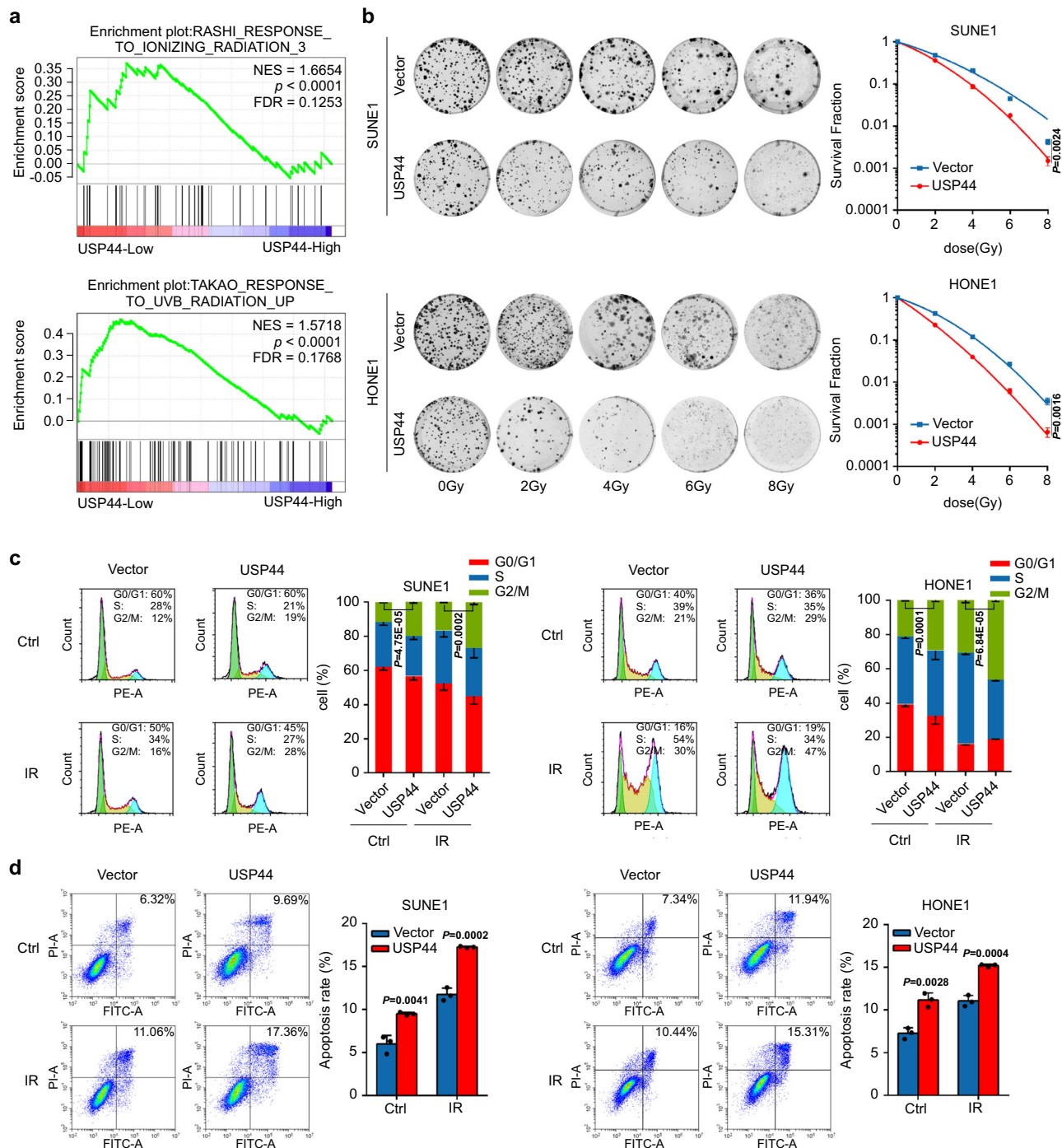

**Fig. 2 USP44 enhances the radiosensitivity of NPC cells in vitro. a** In the GSEA of GSE12452 gene expression data, ionising radiation response-related pathways were enriched in the low *USP44* expression group. **b** Clonogenic assays and survival fraction curves of SUNE1 and HONE1 cells stably transfected with *USP44* or empty vector plasmids after exposure to the indicated IR dose. **c, d** Cell cycle distribution (**c**) and apoptosis rate (**d**) of SUNE1 and HONE1 cells transiently transfected with *USP44* or the empty vector plasmids with or without exposure to 6Gy IR. The cell cycle distribution was detected at 8 h after IR and apoptosis rate was detected at 24 h after IR. Data were presented as the mean ± SD; the *P* values were determined using the two-tailed Student's *t*-test; *n* = 3 independent experiments. Source data are provided as a Source Data file.

degrade Ku80 and thus exerts a radiosensitizing effect, we performed a comet assay to measure DSBs remaining at various times after IR treatment in Vector + sgNC, *USP44* + sgNC and *USP44* + sgTRIM25 grouped SUNE1 or HONE1 cells (Supplementary Fig. 8a). While the levels of DNA damage indicated by comet tails gradually returned to baseline in the Vector + sgNC cells 24 h after IR treatment, it remained higher in the *USP44* + sgNC cells, suggesting there were delays in DNA repair in the

USP44 overexpression cells. Moreover, TRIM25 knockout reversed these DNA damage, suggesting that USP44 has a negative impact on DSB repair by regulating TRIM25 (Fig. 5a). Consistent with the notion that USP44 impedes DSB repair, ectopic expression of USP44 enhanced the formation of DSB marker γH2AX foci induced by IR, which could be reversed by knockout of TRIM25 (Fig. 5b). Furthermore, laser microirradia- tion and live-cell imaging analysis also indicated that USP44

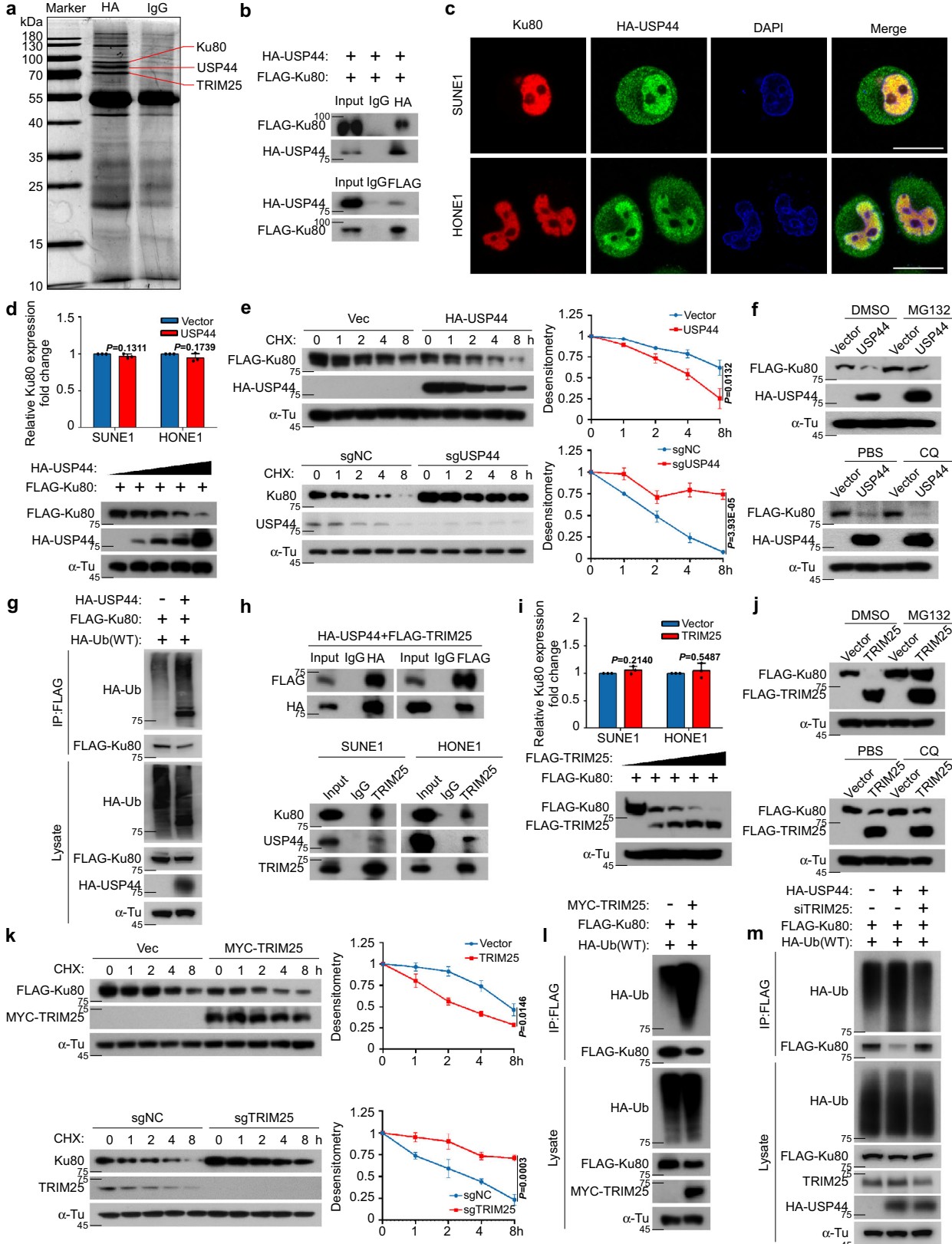

overexpression remarkably resulted in impaired recruitment of GFP-Ku80 at DSB sites, and this effect could be largely reverted by TRIM25 knockout (Fig. 5c). Together, all these observations strongly suggested that the DSB repair activity is impaired by USP44 overexpression, which could be reversed by TRIM25 knockout.

To test whether NHEJ is the pathway affected by USP44, we performed the NHEJ report assay to see the effect of the USP44-TRIM25 axis on the NHEJ repair. As the schematic shows, when the EJ5-GFP plasmids are transfected into NPC cells, GFP will not be produced. While if we infect the cells with the adenoviruses expressing endonuclease I-SceI, the endonuclease I-SceI will recognise

**Fig. 3 USP44 ubiquitinates and degrades Ku80 by recruiting TRIM25. a** SDS-PAGE of HA-immunoprecipitated proteins separated from SUNE1 cells stably overexpressing HA-USP44. Red lines indicate the proteins of interest. **b** Co-IP with anti-HA or anti-FLAG antibody in SUNE1 cells revealed the exogenous association of USP44 and Ku80. **c** Immunofluorescence staining revealed the cellular location of exogenous HA-USP44 (green) and endogenous Ku80 (red) at 0.5 h after exposure to 6Gy IR. Scale bars, 10 μm. **d** USP44 inhibited Ku80 protein expression but not its mRNA expression in a dose-dependent manner. **e** The effect of CHX treatment and greyscale analysis of the results in 293T cells transfected with FLAG-*Ku80* and HA-*USP44* or the empty vector plasmids, as well as in sgNC or sg*USP44* SUNE1 cells. **f** The effect of MG132 and CQ treatment in 293T cells transfected with the indicated plasmids. **g** HEK293T cells transfected with FLAG-*Ku80*, HA-*Ub* and HA-*USP44* or the empty plasmids were subjected to denature-IP and immunoblotted with the indicated antibodies. **h** Co-IP assay detecting the exogenous association of USP44 and TRIM25 and the endogenous association of USP44, TRIM25 and Ku80 in NPC cells. **i** TRIM25 inhibited Ku80 protein expression but not its mRNA expression in a dose-dependent manner. **j** The effect of MG132 and CQ treatment in 293T cells transfected with FLAG-*Ku80* and FLAG-*TRIM25* or the empty vector plasmids. **k** The effect of CHX treatment and greyscale analysis of the results in 293T cells transfected with FLAG-*Ku80* and MYC-*TRIM25* or the empty vector plasmids, as well as in sgNC or sg*TRIM25* SUNE1 cells. **l, m** HEK293T cells transfected with the indicated plasmids or siRNAs were subjected to denature-IP and then immunoblotted with the indicated antibody. Data in **d, e** and **i, k** are presented as the mean ± SD; the *P* values were determined using the two-tailed Student's *t*-test; *n* = 3 independent experiments. Source data are provided as a Source Data file.

and cut the I-SceI sites to produce DSBs, then if the DSBs are repaired through the NHEJ-mediated pathway, the GFP will be restored (Supplementary Fig. 8b)[11,37,38]. As with the results of knockdown of Ku70 (a known essential protein for NHEJ repair)[39,40], overexpression of USP44 significantly decreased GFP expression and thus inhibited NHEJ-mediated DNA repair, and knockout of TRIM25 reversed this inhibitory effect (Fig. 5d and Supplementary Fig. 8c, d). These results demonstrate that USP44 could inhibit NHEJ-mediated DNA repair by targeting TRIM25. More importantly, the suppressive effects on NPC cell survival, proliferation, G2/M phase arrest and apoptosis induced by USP44 overexpression were almost completely recovered by knockdown of TRIM25 (Fig. 6a–d). The suppressive effect of USP44 overexpression on NPC cell survival was also reversed by re-expression of Ku80 (Supplementary Fig. 8e). Overall, these results indicate that the TRIM25-Ku80 axis is a functional target of USP44 that mediates its radiosensitizing effect in NPC.

**USP44 increases the radiosensitivity of NPC cells in vivo.** To determine whether USP44 promotes the radiosensitivity of NPC cells in vivo, we generated subcutaneous tumour xenograft models. Compared with the control group, the USP44 group exhibited reduced xenograft growth in terms of the size, volume, and weight of the excised tumours, especially after IR, indicating that the tumours in the USP44 group were much more sensitive to IR (Fig. 7a–c). The protein levels of TRIM25 and caspase 3, a cell apoptosis-related protein, were increased, and the levels of Ku80 were decreased in the tumours of the USP44 group compared with those of the control group (Fig. 7d, e). Moreover, the radiosensitization effect of USP44 was almost completely rescued by the knockout of TRIM25 in vivo (Fig. 7f–h). These data suggest that overexpression of USP44 regulates TRIM25/Ku80 expression, thus inhibiting cell proliferation and activating cell apoptosis to promote the radiosensitivity of NPC cells in vivo.

**Low expression of USP44 indicates poor prognosis and is associated with tumour relapse.** To further investigate the clinical significance of USP44 protein levels in NPC patients, we conducted IHC staining of 376 NPC tissues with an antibody against USP44. We found positive expression of USP44 in cells from NPC tissues in both the cytoplasm and nucleus, and the samples were grouped according to staining intensity (weak, moderate or strong) (Fig. 8a). We combined these results with the clinical data and found that locoregional recurrence was evidently related to weak USP44 staining in tumour samples (Fig. 8b). We divided these NPC patients into high USP44 expression or low USP44 expression groups for Kaplan–Meier analysis, which revealed significant differences in locoregional recurrence-free,

disease-free and overall survival (Fig. 8c–e). Lower USP44 expression was significantly correlated with a higher risk of relapse, disease and death (Supplementary Table 2). Further analysis identified USP44 expression level, WHO type and TNM stage as independent prognostic indicators for NPC prognosis (Fig. 8f–h). Besides, IHC staining indicated that the USP44 expression was negatively correlated with the Ku80 expression in NPC tissue samples (Supplementary Fig. 9a, b). Taken together, our findings show that low expression of USP44 indicates a poor prognosis and is associated with tumour relapse in NPC patients.

## Discussion

Our current findings demonstrate that the promoter of *USP44* is generally hypermethylated in NPC, which leads to the down-regulation of USP44. USP44 significantly enhances NPC radiosensitivity in vitro and in vivo by stabilising the E3 ligase TRIM25, which further degrades Ku80 via the ubiquitin-proteasome pathway in the NHEJ-mediated DNA repair process. Moreover, we revealed that reduced expression of USP44 indicates a poor prognosis and is associated with tumour relapse in NPC patients.

Radiotherapy is the main treatment regimen for NPC[41–43], but some patients exhibit radioresistance, thus leading to poor therapeutic efficacy and a poor prognosis[44,45]. The role of DNA methylation has been extensively explored in the pathogenesis and development of cancers, including NPC[46]. Several aberrantly methylated genes have been reported as potential prognostic biomarkers for NPC[47–49]. However, the exact regulatory mechanisms remain to be elucidated. Therefore, exploring and clarifying the molecular mechanisms of tumorigenesis and progression caused by DNA methylation is vital for improving the prognosis and providing potential targets for NPC treatment. Hence, we analysed genome-wide DNA methylation between normal human nasopharyngeal samples and NPC tumour samples[24] and found that the CpG islands of *USP44* were frequently hypermethylated in NPC samples; these results were confirmed by analysis of another published microarray dataset and by bisulfite pyrosequencing. TCGA database analysis also revealed that the promoter of *USP44* was broadly hypermethylated in eight other solid tumours. In addition, hypermethylation of the *USP44* promoter was found in colon cancer and breast cancer[17,50], suggesting that the mechanisms of USP44 in suppressing NPC may be shared by other tumours.

DUBs oppose the function of E3 ligases[51,52]. Various DUBs, such as USP7, USP22 and USP28, participate in the complex process of tumorigenesis and progression[53–55]. USP44 has been reported to act as a tumour suppressor that regulates cell cycle arrest and DSB responses by modulating H2B mono-ubiquitylation[20,23]. Our study

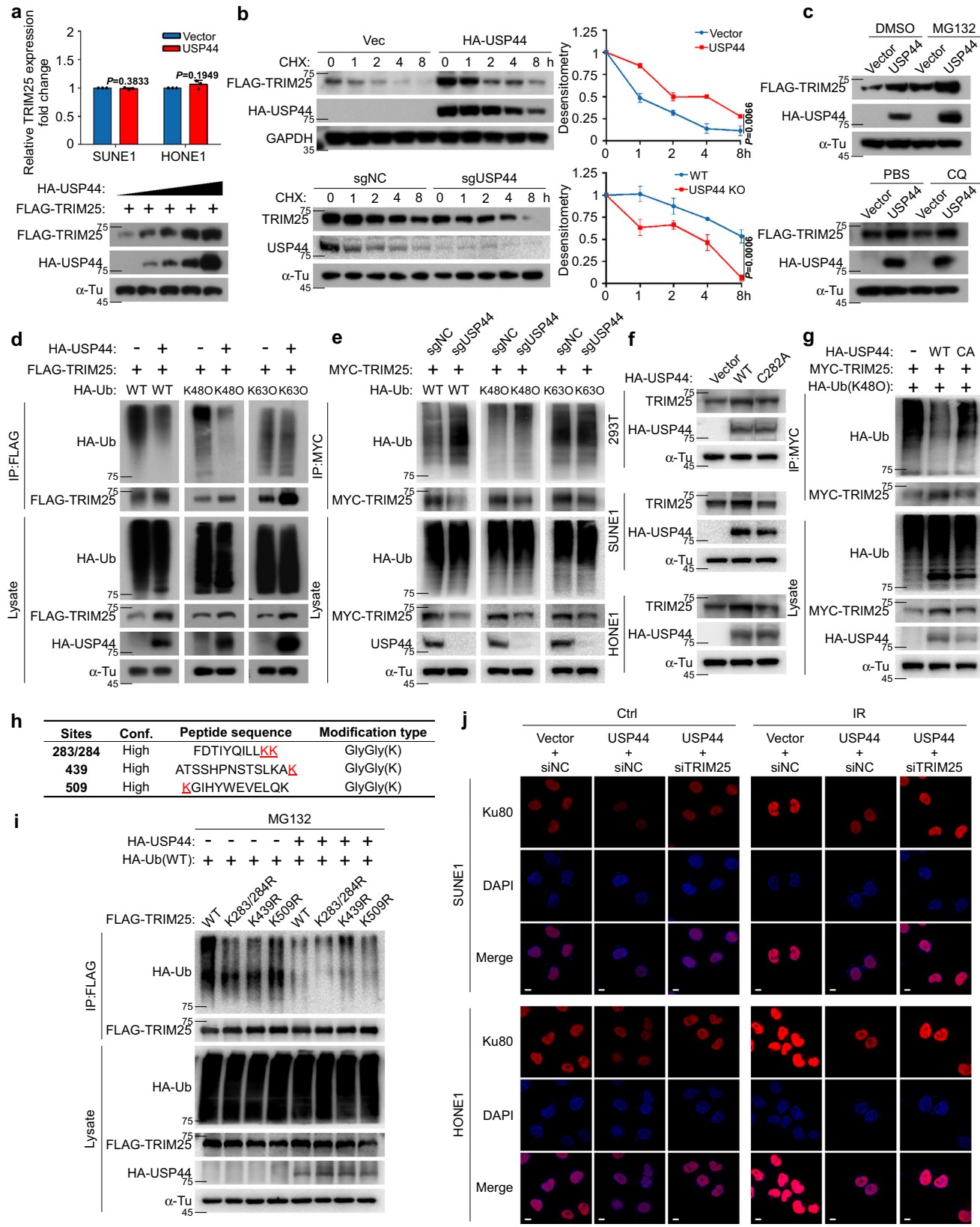

showed that USP44 arrested NPC cells in the G2/M phase indicated by H3S10$_P$ fluorescence. USP44 could also cause G2/M phase arrest by preventing the premature activation of the APC to regulate mitotic checkpoint and binding to the centriole protein centrin to regulate centrosome positioning[20,56]. We found that USP44

promoted G2/M phase arrest, apoptosis induction and radio-sensitization of NPC cells through the TRIM25-Ku80 axis in vivo and in vitro. Our findings uncover an uncovered mechanism by which USP44 regulates the cell cycle and DSB response to induce a tumour suppressor effect in NPC. First, we found that USP44

**Fig. 4 USP44 deubiquitinates and stabilises TRIM25 to promote Ku80 ubiquitination. a** USP44 promoted Ku80 protein expression but not its mRNA expression in a dose-dependent manner. **b, c** The effect of CHX (**b**), MG132 and CQ (**c**) treatment in 293T cells transfected with FLAG-TRIM25 and HA-USP44 or the empty vector plasmids, as well as in sgNC or sgUSP44 SUNE1 cells. **d, e** HEK293T cells transfected with HA-USP44 or the empty vector (**d**) and sgNC or sgUSP44 SUNE1 cells (**e**) co-transfected with FLAG-TRIM25 or MYC-TRIM25 and a vector encoding HA-WT-Ub or its mutants (HA-K48O-Ub or HA-K63O-Ub) were subjected to denature-IP and immunoblotted with the indicated antibodies. **f** HEK293T and NPC cells transfected with vector plasmid, HA-USP44 or HA-USP44 (C282A) were immunoblotted with the indicated antibodies. **g** HONE1 cells transfected with the vector plasmid, HA-USP44 or HA-USP44 (C282A) together with MYC-TRIM25 and HA-K48O-Ub were subjected to denature-IP and immunoblotted with the indicated antibodies. **h** Mass spectrometry analysis of TRIM25 ubiquitination sites. **i** HEK293T cells were transfected with the vector plasmid or HA-USP44, HA-Ub and Flag-TRIM25 WT or KR mutants, subjected to denature-IP with anti-Flag beads and then analysed by immunoblot with an anti-HA or anti-Flag antibody. **j** SUNE1 and HONE1 cells exposed to IR (6Gy) transfected with the indicated plasmids and siRNAs were fixed 0.5 h later and co-immunostained with the anti-Ku80 antibody. Scale bars, 10 μm. Data in **a** and **b** are presented as the mean ± SD; the P values were determined using the two-tailed Student's t-test; n = 3 independent experiments. Source data are provided as a Source Data file.

interacts with the Ku80 protein. Interestingly, we then found that overexpression of USP44 facilitated the degradation of the Ku80 protein rather than stabilising it. We supposed that there might be an intermediate molecule between USP44 and Ku80 that is responsible for the subsequent degradation of Ku80. As expected, we identified an E3 ligase, TRIM25, which acts as a scaffold protein between USP44 and Ku80. It has been reported that TRIM25 interacts with PCNA and p53 in the DNA repair process[57,58]. However, we found that USP44-TRIM25 interaction could degrade Ku80 to inhibit NHEJ-mediated DNA, which combined with G2/M phase arrest and apoptosis induction to subsequently enhance radiosensitivity in NPC. On the one hand, USP44 recruits and stabilises TRIM25 by removing its K48-linked polyubiquitin chain at Lys439. On the other hand, TRIM25 ubiquitinates Ku80 to trigger its degradation through the ubiquitin-proteasome pathway. The ubiquitination of Ku80 is important for its recruitment to and release from DSBs[10,12]. Through laser microirradiation and live-cell imaging analysis, we found that USP44 remarkably impaired the recruitment of Ku80 at DSBs upon laser micro-IR, and this effect could be largely rescued by TRIM25 depletion. This finding reveals an uncovered mechanism by which TRIM25 regulates DSB repair and radiotherapy resistance by targeting Ku80 for ubiquitination.

Although USP44 has been reported as a prognostic indicator in lung cancer, gastric cancer and breast cancer[56,59,60], its prognostic value in NPC remains unknown. We found that the USP44 expression level was significantly correlated with a higher risk of locoregional recurrence, disease progression and death. Low USP44 expression was an independent predictor of poor clinical outcomes in NPC patients. This finding provides a predictive index of curative effects for NPC treatment. In conclusion, Fig. 8i shows our working model. In normal tissues, USP44 recruits and stabilises TRIM25 by removing the K48-linked polyubiquitin chains of TRIM25, and TRIM25 degrades Ku80 by promoting the polyubiquitination of Ku80, which inhibits its recruitment to DSBs and further impairs NHEJ-mediated DNA repair and enhances NPC radiosensitivity. In NPC, hypermethylation of the USP44 promoter leads to its downregulation at the mRNA and protein levels, which blocks the antitumour effect of the USP44-TRIM25-Ku80 axis. Our research has laid a foundation for better understanding the mechanisms of radioresistance.

## Methods

**Clinical specimens**. We collected 19 fresh-frozen NPC specimens and 17 normal nasopharyngeal epithelial specimens, as well as 376 paraffin-embedded locoregionally advanced NPC specimens between January 2006 and December 2009, from Sun Yat-sen University Cancer Center (Guangzhou, China). None of the patients who provided specimens had been treated with anticancer therapies before biopsy. The tumour-node-metastasis (TNM) stages were reclassified according to the 7th edition of the American Joint Committee on Cancer (AJCC) Cancer Staging Manual[61], as the pathological types were classified according to WHO types. All patients underwent radical radiotherapy combined with platinum-based chemotherapy. The clinical features of selected patients are shown in Supplementary Table 2. Our study was

approved by the Institutional Ethical Review Boards of Sun Yat-sen University Cancer Center and the requirement for informed consent was waived by the ethics review boards. This study was conducted according to the REporting recommendations for tumour MARKer prognostic studies (REMARK) guidelines.

**Cell culture**. The immortalised normal human nasopharyngeal epithelial cell line NP69 was cultured in keratinocyte serum-free medium (Invitrogen) supplemented with bovine pituitary extract (BD Biosciences). Human NPC cell lines (SUNE1, CNE1, CNE2, HNE1 and HONE1) were cultured in RPMI-1640 (Invitrogen) medium supplemented with 10% foetal bovine serum (FBS, ExCell Bio). Professor Musheng Zeng (Sun Yat-sen University Cancer Center, China) generously provided nasopharyngeal epithelial cells and NPC cell lines. HEK293T cells obtained from the American Type Tissue Culture Collection (ATCC) were cultured in DMEM (Invitrogen) supplemented with 10% FBS.

**5-Aza-2′-deoxycytidine (DAC), cycloheximide (CHX), MG132 and chloroquine (CQ) treatment**. For the methyltransferase inhibitor DAC (Sigma) treatment, the cells grown for 24 h were treated with or without 10 mM DAC; the drug was replaced every 24 for 72 h. For proteasome inhibitor MG132 (Sigma) and lysosome inhibitor CQ (Sigma-Aldrich) treatment, the transfected cells were treated with 10 μM MG132 or 50 μM CQ for 6 h. For protein synthesis inhibitor CHX (Sigma) treatment, the transfected cells were treated with 100 μg/mL CHX for the indicated time (0, 1, 2, 4, 8 h).

**Plasmid construction and transfection**. The USP44, TRIM25 and Ku80 coding regions were separately tagged with HA, MYC and FLAG and cloned into empty loading plasmids to obtain the overexpression plasmids pSin-EF2-puro-USP44-HA, pSin-EF2-puro-TRIM25-MYC, pSin-EF2-puro-TRIM25-FLAG and pSin-EF2-puro-Ku80-FLAG. The XRCC5 coding regions were cloned into the pEGFPN1 plasmid to obtain the plasmid pEGFPN1-Ku80. In addition, the plasmids pEnter-kana-Ku80-FLAG, pCMV-kana-TRIM25-FLAG, pCMV-kana-Ub (WT)-HA and pEnter-kana-vector plasmids were purchased from Vigene Bioscience (China). PRK-HA-Ub (K48O or K63O) was a gift from Professor Bo Zhong (Wuhan University, China). USP44 shRNA sequences #1 and #2 were obtained according to the shRNA sequence prediction website Portals. The shRNAs were synthesised and cloned into the pLKO.1-RFP vector to obtain PLKO.1-shUSP44 #1/2 plasmids. The TRIM25 siRNA (siTRIM25) was purchased from RiboBio (China). The shRNA and siRNA sequences are listed in Supplementary Table 3.

For transient transfection, the indicated cells were transfected with overexpression plasmids and siRNA using Lipofectamine 3000 (Invitrogen) according to the manufacturer's protocols and harvested 24–48 h post-transfection. The PLKO.1-shUSP44 #1/2 plasmids were transiently transfected into NPC cells with stable USP44 overexpression because of the low expression of USP44 in NPC cells. For stable transfection, HEK293T cells were co-transfected with lentivirus packaging plasmids pMD2.G and pSPAX2 using polyethylenimine (PEI) transfection reagent to collect the virus supernatant. NPC cells were infected with the virus supernatant, screened with puromycin (0.5–1 μg/ml) 48 h post-infection and harvested for RT-qPCR and western blotting assays to determine the expression efficiency of the target gene.

**CRISPR/Cas9-mediated generation of knockout (KO) cells**. The single guide RNA (sgRNA) sequences targeting the USP44 or TRIM25 genomic sequence were designed using an online sgRNA design tool (https://benchling.com) and cloned into pX458 plasmids. These constructs were transfected into cells using Lipofectamine 3000 transfection reagent. The sgUSP44 cells were constructed upon SUNE1 cells with stable USP44 overexpression. Cells with green fluorescence were then sorted with a flow cytometer (MoFlo Astrios) 36 h after transfection, and single colonies were obtained by serial dilution and amplification. Clones were identified by immunoblot with anti-USP44 or anti-TRIM25 antibodies. The sgRNA sequences (5′–3′) are listed in Supplementary Table 3.

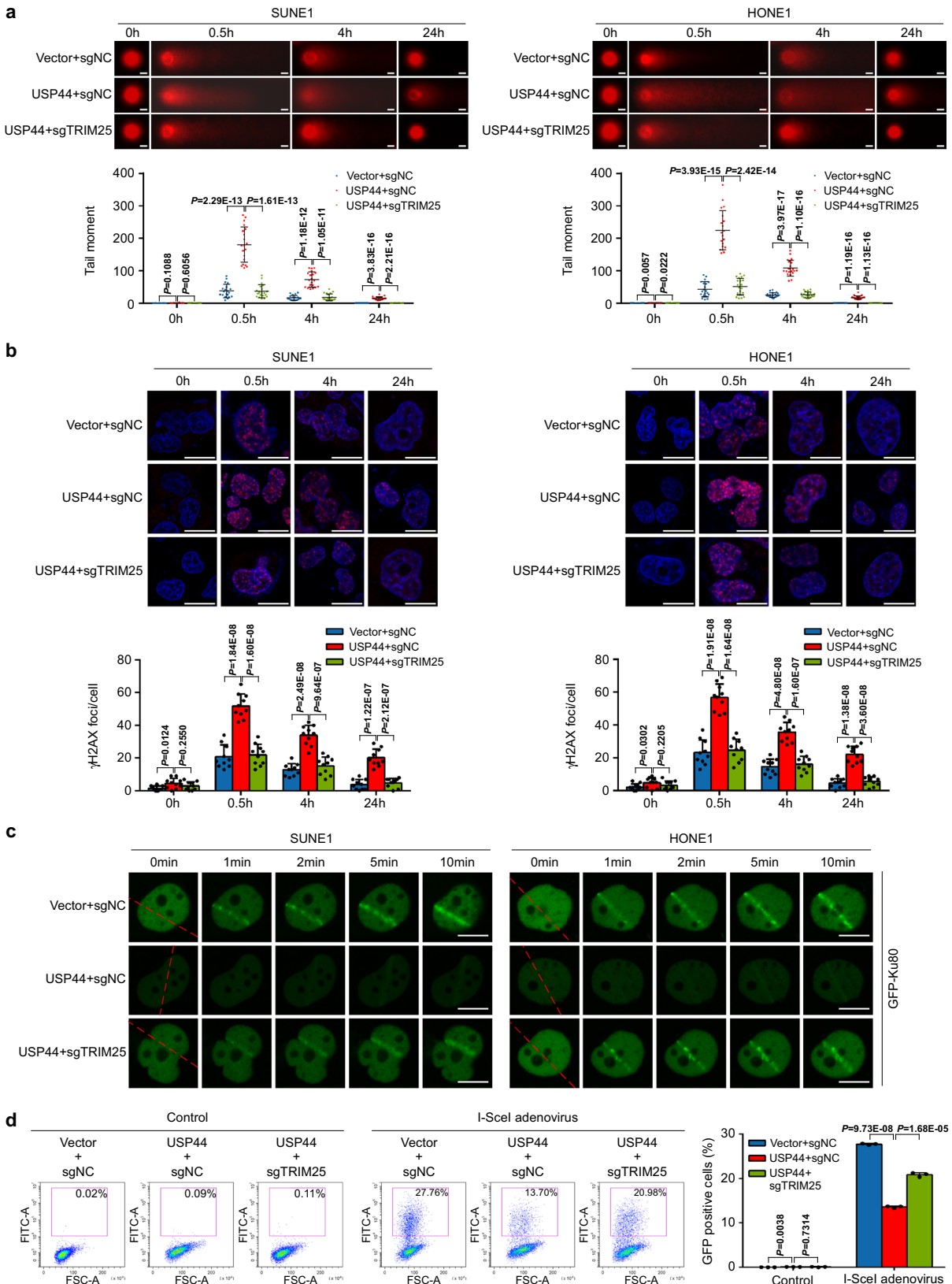

**Bisulfite pyrosequencing analysis**. Fresh-frozen specimens and cell lines were treated with the AllPrep RNA/DNA Mini Kit (Qiagen) or EZ1 DNA Tissue Kit (Qiagen), respectively, to extract genomic DNA. Genomic DNA was modified by bisulfite using the EpiTect Bisulfite Kit (Qiagen). PyroMark Assay Design Software 2.0 (Qiagen) was used to design the *USP44* bisulfite pyrosequencing primer and PCR primer listed in Supplementary Table 3. The sequencing reaction and methylation level quantification were performed with the PyroMark Q96 ID System (Qiagen).

**RT-qPCR assay**. Total RNA from fresh-frozen specimens and cell lines was extracted with the AllPrep RNA/DNA Mini Kit (Qiagen) or TRIzol reagent

**Fig. 5 USP44-TRIM25 increases DSBs by impeding Ku80 recruitment.** The Vector + sgNC, USP44 + sgNC and USP44 + sgTRIM25 SUNE1 or HONE1 cells were stably constructed. **a** Representative comet images and quantitative analysis of tail moments for 6Gy-IR-induced DNA damage in the indicated SUNE1 or HONE1 cells, measured by the comet assay. Scale bars, 10 μm. **b** Representative images and quantitative analysis of the number of γH2AX foci in the indicated SUNE1 and HONE1 cells with or without 6Gy-IR exposure. Scale bars, 10 μm. **c** The indicated SUNE1 and HONE1 cells were transfected with GFP-Ku80 and then subjected to laser micro-IR and live-cell imaging. Scale bars: 10 μm. **d** The indicated SUNE1 cells were transfected with EJ5-GFP, infected with or without I-SceI adenovirus and analysed for GFP positivity by flow cytometry. Data in **a**, **b** and **d** are presented as the mean ± SD; the P values were determined using the two-tailed Student's t-test; n = 20 (**a**), n = 10 (**b**), n = 3 (**d**) repeats from three independent experiments. Source data are provided as a Source Data file.

(Invitrogen). Complimentary DNA was produced using random primers and M-MLV reverse transcriptase (Promega). RT-qPCR was performed using SYBR Green PCR master mix (Applied Biosystems) and a CFX96 Touch sequence detection system (Bio-Rad). Relative gene expression was calculated by the 2-ΔΔCT equation with GAPDH as an internal control. The primer sequences used for the RT-qPCR assay are listed in Supplementary Table 3.

**Western blot assay**. Fresh-frozen specimens were ground in liquid nitrogen and lysed to obtain total protein. Cell lines were lysed and sonicated to obtain total protein. Total protein was separated by SDS-PAGE (Genscript) and transferred to PVDF membranes (Millipore). The membranes were blocked in 5% skim milk and incubated overnight with primary antibodies. Following incubation with HRP-linked secondary antibodies, the bands of interest were detected by the X-ray film method. The antibodies used are listed in Supplementary Table 4. Unprocessed scans of immunoblots are provided as Supplementary Fig. 10.

**Cell viability assay**. The cells were plated into 96-well plates at densities of 800 HONE1 cells or 1000 SUNE1 cells per well. On the indicated days (days 0, 1, 2, 3 and 4), 10 μl Cell Counting Kit-8 (CCK-8) reagent (Dojindo) per well was added to the 96-well plates. After incubation at 37 °C for 2 h, the absorbance of each well at 450 nm was detected on a spectrophotometer.

**Clonogenic assay**. Single-cell suspensions were inoculated into 6-well plates (200–10,000 cells per well) and treated with IR (0–8 Gy) until cell adherence. After colony formation (~10–14 days), the plates were rinsed with PBS, fixed with methanol and stained with crystal violet. Colonies containing more than 50 cells were counted. Cell survival curves were fitted according to the linear-quadratic (LQ) formula: surviving fraction $(SF) = exp\ (-\alpha D - \beta D^2)$.

**Flow cytometry analysis of cell cycle and apoptosis**. The Cell Cycle and Apoptosis Kit (Keygen Biotech) was applied to detect the cell cycle distribution and apoptosis rate of each sample. For cell cycle analysis, serum-starved cells were collected 8 h after 6Gy IR or no IR, washed in PBS and fixed in 70% ice-cold ethanol overnight. After washing, each sample was stained with 500 μl RNase A: PI (1:9, v/v) dyeing solution and screened. The cell cycle distribution was detected using an ACEA NovoCyte flow cytometer and analysed with NovoExpress 1.3.0 software. For apoptosis analysis, cells were collected 24 h after 6Gy IR or no IR and washed twice with PBS. Each sample was resuspended in 500 μl binding buffer, screened and incubated with 5 μl Annexin V-FITC and 5 μl PI fluorescent dyes. The apoptosis rate was detected using a cytoFLEX flow cytometer and analysed with CytExpert 2.2 software. FITC−/PI− cells were considered viable cells, FITC+/PI− cells were considered early apoptotic cells and FITC+/PI+ cells were considered late apoptotic or dead cells. The gating strategy is provided in Supplementary Fig. 11.

**Mass spectrometry and co-immunoprecipitation (co-IP) assay**. For the IP assay, cells were lysed on ice with IP lysis buffer and sonicated. Total protein was immunoprecipitated overnight at 4 °C with 3 μg of the indicated antibodies. The immune complexes were added to Pierce™ Protein A/G Magnetic Beads (Thermo Scientific) and then washed with IP Wash Buffer. The collected immune complexes were separated by SDS-PAGE and stained with Coomassie blue. Mass spectrometry was performed by Huijun Biotechnology (China). The proteins of interest in the Co-IP were detected by western blot assay. The antibodies used are listed in Supplementary Table 4.

**Immunofluorescence and confocal microscopy**. Cells were fixed in 0.4% paraformaldehyde, permeabilized in 0.5% Triton X-100, blocked in 1% BSA-PBS and incubated overnight at 4 °C with primary antibodies. The coverslips were then stained with secondary antibodies, stained with 4′,6-diamidino-2-phenylindole (DAPI, Sigma) and sealed to prevent quenching. Fluorescence images were captured using a confocal scanning microscope (LSM880 with Fast Airyscan, ZEISS). The antibodies used for immunofluorescence are listed in Supplementary Table 4.

**Denature-IP assay**. All the ubiquitin assays were performed in denaturing conditions. The denature-IP assays were done according to the methods in previous papers[62,63]. At 24 h post-transfection, cells were lysed on ice in NP-40-containing lysis buffer supplemented with EDTA-free protease inhibitor cocktail (Roche). Cell lysates were denatured at 95 °C for 5 min in the presence of 1% SDS, and immunoprecipitated according to the methods of co-IP with specific antibodies. The lysates and immune complexes were subjected to western blot analysis.

**Comet assay**. Cells were exposed to IR (6Gy) and harvested at the indicated time points after IR. Neutral comet assays were conducted with the SCGE DNA Damage Detection Kit (KeyGentec) and stained with propidium iodide (PI). The quantitation of tail moments was analysed with CaspLab-Comet Assay Software, and 20 cells were scored for each case.

**Laser microirradiation**. DSBs were generated with a UVA laser using a pulsed sub-cell illumination system under a 60 × objective lens for live-cell imaging. Cells were seeded on 35-mm glass-bottom dishes (Nest, China), transiently transfected with GFP-Ku80 overnight, visualised 24 h after transfection with a Nikon AX confocal microscope and micro-irradiated with a λ = 365 nm, 16 Hz pulse, 65% energy UVA laser of a Micropoint Ablation System (Andor, USA). Consecutive images were captured at 10 s interval for 10 min.

**NHEJ reporter assay**. The EJ5-GFP plasmid was generously provided by Professor Muyan Cai (Sun Yat-sen University Cancer Center, China). The NHEJ reporter assay was performed according to the methods as previously described[64]. Cells were seeded in 6-well plates, transfected with EJ5-GFP and infected with I-SceI-expressing adenovirus after 18 h. The medium was replaced after 14 h to avoid adenovirus toxicity. Cells were harvested after 72 h, and the percentage of GFP-positive cells was quantitated by flow cytometry to assess NHEJ-mediated DNA repair efficiency. The gating strategy is provided in Supplementary Fig. 11.

**Murine xenograft growth of NPC**. Female BALB/c nude mice (6–8 weeks old; n = 64) were purchased from Charles River Laboratories (Beijing, China) and housed in barrier facilities on a 12 h light/dark cycle at temperature 18–22 °C and humidity 50–60%. Mice were randomly assigned for tumour injection and administered a subcutaneous injection of $3 \times 10^5$ SUNE1 cells. The tumour volume and bodyweight of the injected mice were measured every three days from day seven after tumour injection. After the diameter of xenograft tumours reached ~5 mm, the mice were locally irradiated with 8Gy once, while the control mice were not irradiated. Tumour volume was calculated using the following formula: length × width$^2$ × 0.5. After 28 days, tumour samples were paraffin-embedded for immunohistochemistry (IHC) analyses. All experimental protocols were approved by the Institutional Animal Care and Use Committee of Sun Yat-sen University and complied with the Declaration of Helsinki. We did our best to minimise animal suffering. The maximal tumour diameter was 20 mm permitted by our ethics committee, and the maximal tumours size was not exceeded in our study.

**IHC**. Sections were deparaffinized, rehydrated, preincubated with hydrogen peroxide, blocked with goat serum (Beyotime), incubated with primary antibodies and labelled with HRP rabbit/mouse secondary antibodies (Dako REAL™ EnVision™), stained with diaminobenzidine (Sigma) and counterstained with haematoxylin. Images were obtained with an AxioVision Rel.4.6 computerised image analysis system (Carl Zeiss). All sections were scored by two experienced pathologists according to the immunoreactive score (IRS) system[65]. The intensity of staining was scored as follows: 0 (negative staining), 1 (weak staining), 2 (moderate staining) and 3 (strong staining). The percentage of positive tumour cells was scored as follows: 1 (<10%), 2 (10–35%), 3 (35–70%) and 4 (>70%). The IRS scores were calculated as the product of the staining intensity score and the score of percentage of positive tumour cells. The antibodies used in the IHC analysis are listed in Supplementary Table 4.

**Statistics and reproducibility**. All statistical analyses were performed using the SPSS version 22.0 statistical software and Graph-Pad Prism version 6.0.1 software. Differences between groups were analysed using unpaired two-tailed Student's t-tests or χ²

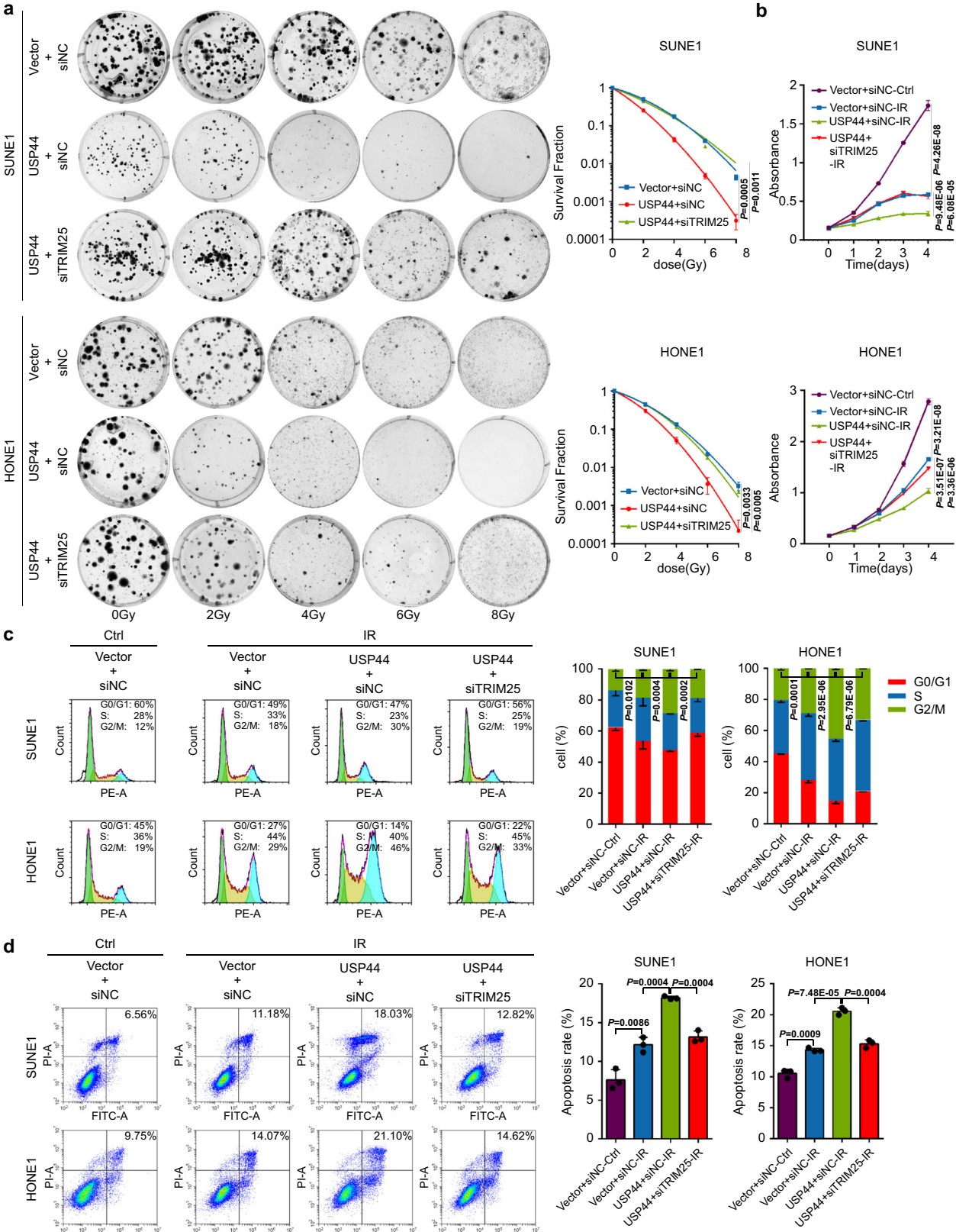

**Fig. 6 Knockdown of TRIM25 reverses the radiosensitizing effect of USP44 in vitro.** SUNE1 and HONE1 cells were transiently co-transfected with HA-*USP44* or the empty vector plasmids plus si*TRIM25* or control siRNA. **a** Clonogenic assays and survival fraction curve analysis (**a**) and CCK-8 assay (**b**) of transfected SUNE1 and HONE1 cells after exposure to the indicated IR dose. The absorbance at 450 nm in **c** is presented as the mean ± SD of *n* = 4 independent experiments. **c, d** Cell cycle distribution (**c**) and apoptosis rate (**d**) of SUNE1 and HONE1 cells with or without exposure to 6Gy-IR. The cell cycle distribution was detected at 8 h after IR and apoptosis rate was detected at 24 h after IR. Data were presented as the mean ± SD; the *P* values were determined using the two-tailed Student's *t*-test; *n* = 3 independent experiments. Source data are provided as a Source Data file.

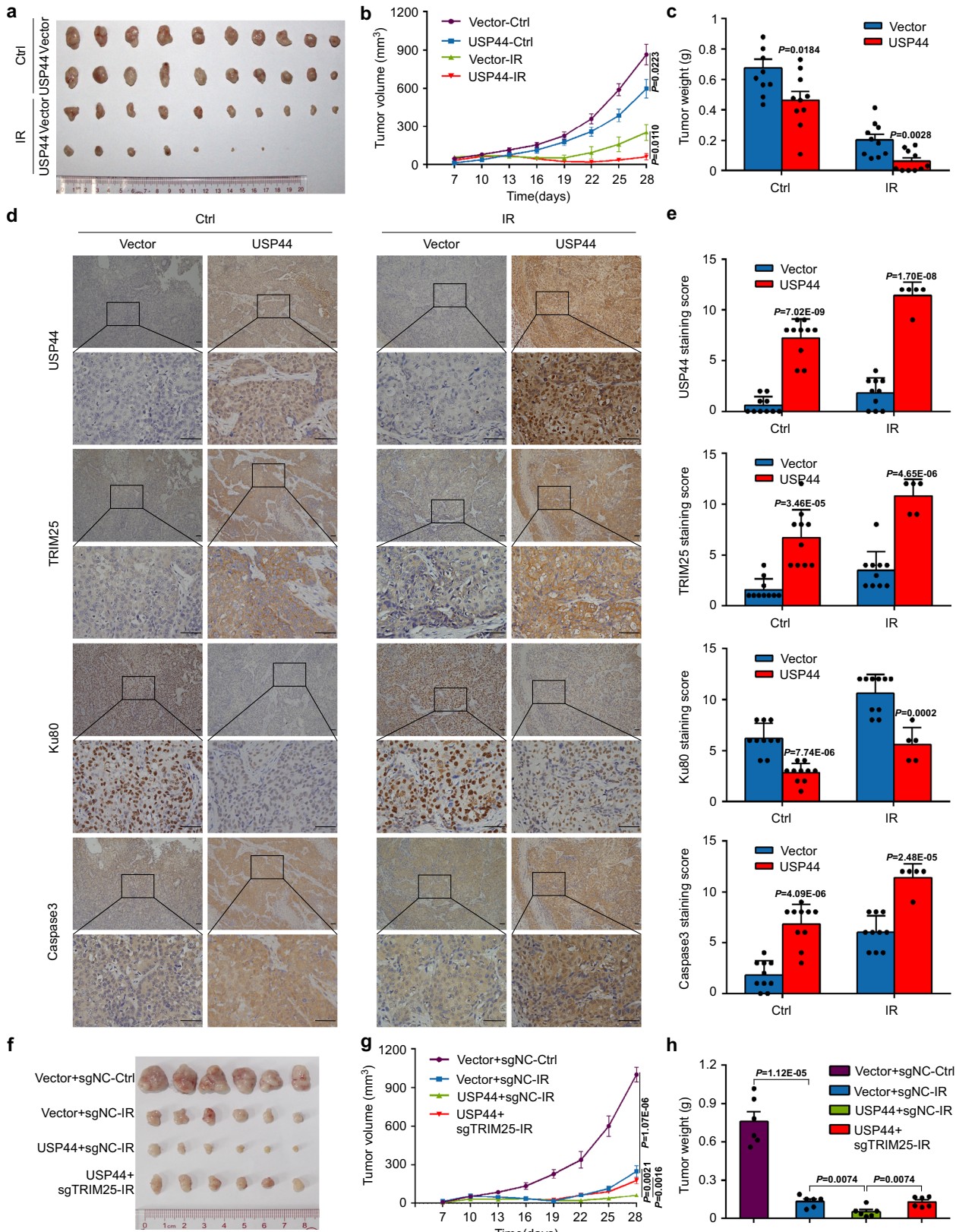

**Fig. 7 USP44 promotes the radiosensitivity of NPC cells in vivo.** The SUNE1 cells stably transfected with indicated plasmids were implanted subcutaneously into female BALB/c nude mice to construct xenograft growth models and exposed to 8Gy IR or not. **a–c** Macroscopic images (**a**), average volume (**b**) and average weight (**c**) of the excised tumours for each group (n = 10). **d**, **e** Representative images of immunohistochemical staining and IHC scores for USP44, TRIM25, Ku80 and caspase 3 expression in the excised tumours from each group (n = 5 for IR+USP44 group and n = 10 for the other three groups). Scale bars, 50 μm. **f–h** Macroscopic images (**f**), average volume (**g**) and average weight (**h**) of the excised tumours for each group (n = 6). Data in **b**, **c**, **g**, **h** are presented as the mean ± SEM, and those in **e** are presented as the mean ± SD; the P values were determined using the two-tailed Student's t-test. Source data are provided as a Source Data file.

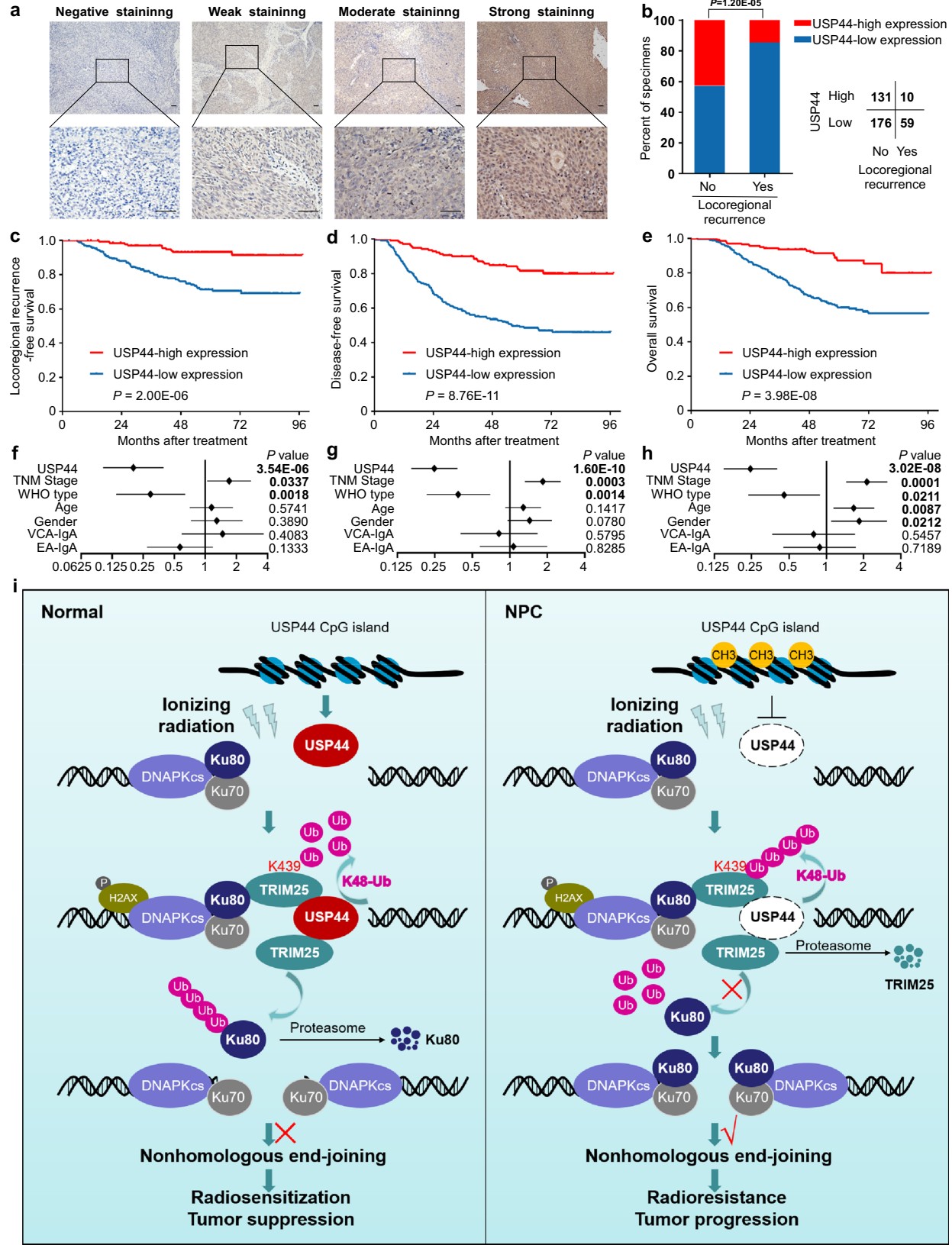

test. Data were presented as the mean ± SD or mean ± SEM and $P < 0.05$ was considered significant. Survival curves were constructed using the Kaplan–Meier method, and the differences among groups were compared by the log-rank test. Multivariate analysis with a Cox proportional hazards regression model was used to determine

independent prognostic factors. The strength of the relationship is evaluated using the Spearman correlation. Unless otherwise indicated, the experiments were performed independently in triplicate, and $n$ is indicated in the figure legends.

**Fig. 8 Low expression of USP44 indicates a poor prognosis and is associated with tumour relapse in NPC patients. a** Representative images of immunohistochemical staining for USP44 protein expression is graded according to the intensity of staining in 376 NPC tissues. Scale bars, 50 μm. **b** Correlations of locoregional recurrence status with the level of USP44 expression detected by IHC. The $P$ value was determined using the two-tailed $\chi^2$ test. **c–e** Kaplan–Meier analysis of locoregional recurrence-free survival (**c**), disease-free survival (**d**) and overall survival (**e**) according to the USP44 expression levels. The $P$ values in **c–e** were determined using the log-rank test. **f–h** Forest plots of multivariate Cox regression analyses showing the significance of different prognostic variables in NPC locoregional recurrence-free survival (**f**), disease-free survival (**g**) and overall survival (**h**). **i** Proposed working model of USP44. USP44 recruits and stabilises TRIM25 by removing the K48-linked polyubiquitin chains of TRIM25, and TRIM25 degrades Ku80 by promoting its polyubiquitination and inhibits its recruitment to DSBs, which further inhibits the NHEJ pathway and enhances NPC radiosensitivity. In NPC, hypermethylation of the *USP44* promoter leads to its downregulation at the mRNA and protein levels, which blocks the anticancer effect of the USP44-TRIM25-Ku80 axis. Source data are provided as a Source Data file.

**Reporting Summary**. Further information on research design is available in the Nature Research Reporting Summary linked to this article.

## Data availability

The microarray data used in this study are available in the Gene Expression Omnibus (GEO; http://www.ncbi.nlm.nih.gov/geo/)under accession codes GSE12452, GSE52068 and GSE62336. The data used in this study for gene expression profiling interactive analysis (GEPIA; http://gepia.cancer-pku.cn/index.html) are available in The Cancer Genome Atlas (TCGA; https://tcga-data.nci.nih.gov/). All the other data supporting the findings of this study are available within the article and its Supplementary Information files. The key raw data have been deposited to Research Data Deposit public platform (https://www.researchdata.org.cn/), with an approval number of RDDB2021760690. Source data are provided with this paper.

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

## Acknowledgements

The authors thank Prof. M. Zeng, Prof. B. Zhong and Prof. M. Cai for cells, plasmids and technical advice. This work was sponsored by the National Natural Science Foundation of China (81922057 to NL; 81572658 to JM), Natural Science Foundation of Guangdong Province (2018B030306045 to NL; 2017A030312003 to JM) and Key-Area Research and Development Programme of Guangdong Province (2019B020230002 to JM).

## Author contributions

Y.C. Y.Z. and X.Y. designed, carried out and analysed most of the experiments. X.R., S.H., S.G. and X.T. contributed to several experiments. J.L., S.H., Y.L., X.H., Q.L., C.D., X.F. and J.M. helped with the data analyses and were involved in discussions of the data. Y.C., Y.Z. and N.L. wrote the manuscript with input from all other authors. N.L. and J.M. conceived the study, provided scientific directions and established collaborations.

## Competing interests

The authors declare no competing interests.
