## [Peer Review File · Nature Communications]

USP44 regulates irradiation-induced DNA double-strand break repair and suppresses tumorigenesis in nasopharyngeal carcinomaREVIEWER COMMENTS

Reviewer #1 (Remarks to the Author):

Ubiquitin-specific protease 44 (USP44) has been implicated in several critical cellular processes, from regulating the spindle assembly checkpoint and chromosome segregation (through cdc20 stabilization) to modulation of gene expression by histone deubiquitination.

The role of this deubiquitinating enzyme (DUB) in cancer, however, is not fully understood, and its functions appear to be cell and context specific. Multiple reports demonstrate that USP44 expression is lost or greatly diminished in several cancers, defining this DUB as a tumor suppressor. On the other hand, high expression of USP44 is associated with T-cell leukemia, and USP44 positive cancer stem cells were found to contribute to cancer aggressiveness in breast cancer. Thus, elucidating the molecular mechanisms linking USP44 with cancer development is important and will likely inform future anti-cancer therapies.

In this study, Chen et al. show that USP44 functions are required for proper DSBs repair (NHEJ) by modulating the Ku80 steady-state cell levels. Mechanistically, USP44 interacts with and stabilizes TRIM25, an E3 ubiquitin ligase, involved in the Ku80 ubiquitination and degradation. The authors demonstrate that the USP44 locus is hypermethylated in nasopharyngeal carcinoma (NPC). This epigenetic modification severely diminishes the expression of USP44 in tumors compared to normal tissue. Furthermore, the reduced levels of USP44 in tumors result in less stable TRIM25 hence higher Ku80 levels, rendering tumors less sensitive to radiotherapy. Notably, the authors demonstrated that reduced USP44 expression negatively correlates with patient survival after treatment.

The authors performed a comprehensive study, presenting high-quality data to support their claims.

Although the presented data are in line with the authors' overall claims, the interpretation of some of the presented results raises concerns. These concerns need to be addressed before this work is considered for publication in Nature Communications.

Major concerns:

1) Based on the results presented in Fig.2b, the authors concluded that overexpression of USP44 severely impacts the (growth) colony formation of NPC cells after irradiation. It appears, however, that USP44 expression impacts the (growth) colony formation (in both cell lines used) regardless of IR. A rough comparison of the colony number/size shows approximately 30-40% fewer colonies in the USP44 lines even before the treatment (0 Gy). Thus, although introducing DNA damage by IR may exacerbate this effect, it does not appear to be the primary reason for the growth defects in observed USP44 expressing NPC cell lines.

2) In Fig 2c, the authors claim that the combination of IR and USP44 overexpression leads to G2M-phase arrest indicating an essential role of USP44 in DNA damage response. It is not mentioned, however, at what time after irradiation the cell cycle was examined. Examining the cell cycle at different time points after irradiation will give a more accurate representation of possible cell cycle defects associated with persisting DSBs in the USP44 expressing cells.

3) It is not mentioned what conditions were used for the IP/IB experiments used to detect the increased ubiquitination of Ku80 and TRIM25 (Figs 3g and 4d). The authors should perform the experiments shown in these figures in denaturing conditions to confirm that the ubiquitinated species precipitated with FLAG/MYC IPs are polyubiquitinated Ku80 and TRIM25. Blots for ubiquitin (HA) can often detect the increased ubiquitination of associated proteins precipitated with the bait proteins. This concern is supported by the observation that there is a vast difference of the HA-Ub level in the first lanes of the HA blots in Figs 3l and 3m, although these should look similar. At least, the authors can perform these experiments in reverse order, where they precipitate ubiquitinated species and blot for Ku80 or TRIM25 to detect the higher molecular weight, ubiquitinated forms of these proteins in the described conditions.

4) Based on the experiments presented in Fig 4i, the authors concluded that USP44 removes ubiquitin specifically from K439 on TRIM25, and by this, inhibits its degradation. However, there might be several other explanations for the results presented in this figure. First, just like for the WT, the authors should examine the ubiquitination level of the other two mutants (K283/284R and K509R) in cells without USP44 expression, i.e., compare cells lysates with or without USP44 expression for all constructs, not just for the WT. As presented now, it is unknown if the reduced ubiquitination of TRIM25-K283/284R and TRIM25-K509R in the blot results from the USP44 activity or if these mutants cannot be ubiquitinated to the same extent as WT and K439R? Second, it is obvious that multiple lysines in TRIM25 can be ubiquitinated and that the K439R mutant displays almost WT levels of ubiquitination. Thus, it is unknown how much K439 contributes to the total ubiquitination of TRIM25 hence its stability in the cells. Furthermore, K439R may introduce structural changes in TRIM25, which prevent its interaction with USP44 and thus its deubiquitination by this DUB. Therefore, the interaction between TRIM25-K439R and USP44 should be tested.

5) In Fig 4j, the authors used IF to illustrate that overexpression of USP44 leads to reduced levels of Ku80 in cells and that this effect can be rescued by TRIM25 ablation. The results in this figure, however, are puzzling. It is unclear why IR will induce such a drastic difference in the Ku80 levels in the cells (comparing the Ku80 signal in Ctrl and IR panels). The total levels of Ku80 in cells are usually not affected by IR (which is also supported by the results presented in Fig 5c). As shown, it appears that USP44 affects the Ku80 induction/expression upon irradiation rather than its stability in these cells. Furthermore, the Ku80 signal seems to be the strongest in USP44 overexpressing cells before IR, which contradicts the data presented in the figures before (Figs 3d and 3e).

6) In Fig 5, the authors present evidence for DNA repair defects in cells overexpressing USP44. However, the data shown in Fig 5a and Fig 5b suggest that the USP44 overexpressing cells are more susceptible to DSBs upon IR rather than have DSBs repair defects. There is no explanation why USP44 expressing cells will form approximately four times longer comet tails 30 min after the same dose of irradiation? It looks like that most of the damage is then repaired in 24 h. The same is true for the γ H2AX foci experiment (Fig 5b), where the USP44 overexpressing cells form more foci than vector-only cells. There is no indication at what time after the irradiation the cells were stained for γ H2AX. To claim that USP44 overexpression impairs DSBs repair, the authors should perform a time kinetic where they compare the changes in the tail moment or foci number over time, starting with comparable amount foci/tail lengths in all cell lines.

7) The data in Fig 5c will argue that the USP44 functions are not as crucial for the total Ku80 levels in the cells but rather for loading on DSBs upon damage. The levels of GFP-Ku80 in HONE1 cells are not significantly different in USP44 overexpressing cells compared to the Vector only and USP44+TRIM25KO cells, yet they failed to form laser-induced Ku80 foci/streaks.

8) It will be important to see if higher levels of USP44 correlate with low levels of Ku80 in patients (Fig 8a).

Minor concerns:

1) There is a discrepancy in the text and the label of Fig 5a. The text mentions that the tail moment remains higher in USP44 overexpressing cells for four hours, whereas the last point in the figure is 24h.

2) There is no evidence that USP44 binds DNA/chromatin (directly); it will be more accurate if the authors depict it not as a DNA-bound protein in their model figure.

Reviewer #2 (Remarks to the Author):

Summary:

Chen et al present data in nasopharyngeal carcinoma (NPC) that expression of the protein USP44 regulates the sensitivity of NPC to radiotherapy. Through analysis of a previously published data set they show that USP44 promoter is hypermethylated in NPC relative to normal epithelium and these results are corroborated in pre-clinical cell line models. They then show that overexpression of USP44 results in increased radiosensitivity and conversely that depletion of USP44 via RNAi leads to relative radioresistance, reportedly through modulation of apoptosis, cell cycle arrest and changes in DNA damage levels. The mechanism is reported to involve the interaction between USP44, TRIM25 and Ku80 (a known regulator of the DNA damage response/NHEJ). Finally their results are demonstrated in an in vivo animal model and with analysis.

Comments:

Overall the data are interesting and are in line with prior results suggesting that USP44 is involved in DNA Repair and recruitment of DNA repair factors such as 53BP1 to sites of DSBs (Ref 23). One significant weak point of this manuscript is the translational/therapeutic relevance of these findings. The authors do not report upon (or even discuss) methods by which these findings may be exploited for therapeutic gain in patients with NPC. Along these lines, generally there is the expectation that a target should be tumor specific (so as to not increase normal tissue toxicity), and the authors have not shown that modulation of USP44 in normal tissues would not lead to increased sensitivity to radiotherapy or other DNA damaging agents.

While the authors have utilized overexpression experiments to show that USP44 is responsible for modulating radiation sensitivity, they have utilized RNAi based techniques to deplete USP44 expression. RNAi techniques are well known to have substantial off-target effects with regards to DNA repair (PMID 22344029) processes and as such use of CRISPR based techniques should be done to confirm that depletion of USP44 modulates radiation survival. In these same experiments the western blots in Suppl Figure 2b do not show convincing changes in USP44 expression between shControl and shUSP44 expression. Furthermore expression in the WT, USP44 knockdown cells and USP44 overexpressed cells should be compared on the same western blot to understand the differences in expression required for changes in radiation response.

Differences attributed to survival we thought to be due to changes in Ku80 driven by USP44 and TRIM25 interaction, however no experiments were done to determine whether this is true (i.e. through re-expression of Ku80 in USP44 overexpressing cells).

There are no statistical tests done to compare cell cycle profiles seen in several figures.

Given the known role of USP44 and regulation of mitosis it would be interesting to understand whether the G2/M checkpoint was altered (e.g. using phosphor Histone H3 staining to quantify mitotic cells) in their experimental models. Additionally, the authors should attempt to discuss/connect the impact of USP44 expression leading to changes in Ku80 expression, NHEJ efficiency and the results seen in Figure 2 (apoptosis and cell cycle arrest).

It is unclear as to the functional significance/magnitude of change of the NHEJ assay changes seen in Figure 5d. This would be clarified by the addition of proper controls to this assay (e.g. including a knockdown of a known essential NHEJ protein).

There should be quantitation of the changes in Figure 4J as well as western blot analysis of Ku80 expression changes with these experimental manipulations.

The radiation dose and timepoints should be specified explicitly throughout the paper. For instance I do not see this specified for figure 5b.

Reviewer #3 (Remarks to the Author):

To increase the efficacy of radiotherapy in the treatment of nasopharyngeal cancer, it is essential to understand the mechanism underlying resistance to radiation in cancer. In this manuscript, the authors show that the novel USP44-TRIM25-Ku80 axis plays a critical role in radiation resistance frequently observed in nasopharyngeal cancer. In general, the experiments are well designed, and the results are properly presented. If the following points are addressed, the manuscript can be published in Nature Communications.

Major points

1. The effect of USP44 and TRIM25 on Ku80 expression is mainly investigated using exogenously expressing cells. The change in endogenous Ku80 expression in SUNE1 and HONE1 cells is only demonstrated by immunofluorescence. If western blot data showing the change at endogenous levels are available, the result could be confirmed.
2. As shown in Suppl Fig 2, USP44 KD induces radiation resistance in SUNE1 and HONE1 cells. If Ku80 plays a direct role in radiation resistance, does silencing of Ku80 cancel radiation resistance in these cells? Alternatively, the same experiment can be done using USP44 KO cells.
3. As shown in Fig. 7, both endogenous and exogenous expression levels of USP44 and TRIM25 are higher in irradiated cells than in non-irradiated cells even at 28 days, indicating that some mechanisms responding to radiation maintain their levels after irradiation. Is there any evidence suggesting that their levels are regulated by post-translational modifications associated with the DNA damage response? If not, this point should be discussed.

Minor points

1. Figure 3c. Colocalization of Ku80 with HA-USP44 should be more clearly demonstrated.
2. Figure 3e & Figure 4b. Why are USP44 bands visible in USP44 KO cells?
3. Suppl Figure 2a. Is HA-USP44 correct in western blot? It may be USP44 to show the level in vector-transfected cells.
4. Reference 11. The page number is missing.

Dear Editor and Reviewers,

Thank you very much for your important and supportive comments, questions and suggestions, which have greatly helped us to improve our study. Enclosed is the revised version of manuscript (Ref.: NCOMMS-21-18892) entitled, “**USP44 regulates irradiation-induced DNA double-strand break repair and suppresses tumorigenesis in nasopharyngeal carcinoma**”. We have revised and modified the manuscript in accordance with the editor and the reviewers’ comments and re-submitted the revised manuscript. We all appreciate your support and efforts on our manuscript and look forward to your further decision, and we sincerely hope to have the opportunity to publish this paper in *Nature Communications*.

A copy of the manuscript indicating where revisions have been made is included in the resubmission. The revisions are indicated using the "Track Changes" function in Word.

The following is the point-by-point response to the reviewers’ comments and questions.

Reviewer #1 (Remarks to the Author):

Ubiquitin-specific protease 44 (USP44) has been implicated in several critical cellular processes, from regulating the spindle assembly checkpoint and chromosome segregation (through cdc20 stabilization) to modulation of gene expression by histone deubiquitination.

The role of this deubiquitinating enzyme (DUB) in cancer, however, is not fully understood, and its functions appear to be cell and context specific. Multiple reports demonstrate that USP44 expression is lost or greatly diminished in several cancers, defining this DUB as a tumor suppressor. On the other hand, high expression of USP44 is associated with T-cell leukemia, and USP44 positive cancer stem cells were found to contribute to cancer aggressiveness in breast cancer. Thus, elucidating the molecular mechanisms linking USP44 with cancer development is important and will likely inform future anti-cancer therapies.

In this study, Chen et al. show that USP44 functions are required for proper DSBs repair (NHEJ) by modulating the Ku80 steady-state cell levels. Mechanistically, USP44 interacts with and stabilizes TRIM25, an E3 ubiquitin ligase, involved in the Ku80 ubiquitination and degradation. The authors demonstrate that the USP44 locus is hypermethylated in nasopharyngeal carcinoma (NPC). This epigenetic modification severely diminishes the

expression of USP44 in tumors compared to normal tissue. Furthermore, the reduced levels of USP44 in tumors result in less stable TRIM25 hence higher Ku80 levels, rendering tumors less sensitive to radiotherapy. Notably, the authors demonstrated that reduced USP44 expression negatively correlates with patient survival after treatment.

The authors performed a comprehensive study, presenting high-quality data to support their claims.

Although the presented data are in line with the authors' overall claims, the interpretation of some of the presented results raises concerns. These concerns need to be addressed before this work is considered for publication in Nature Communications.

Major concerns:

1. Based on the results presented in Fig.2b, the authors concluded that overexpression of USP44 severely impacts the (growth) colony formation of NPC cells after irradiation. It appears, however, that USP44 expression impacts the (growth) colony formation (in both cell lines used) regardless of IR. A rough comparison of the colony number/size shows approximately 30-40% fewer colonies in the USP44 lines even before the treatment (0 Gy). Thus, although introducing DNA damage by IR may exacerbate this effect, it does not appear to be the primary reason for the growth defects in observed USP44 expressing NPC cell lines.

Response:

Thanks for your valuable comments. In our present study, we found that USP44 expression is obviously downregulated in NPC, and GSEA analysis showed that the dysregulation of USP44 remarkably enriched in gene set related to radiation response pathways. Radiotherapy is the primary therapeutic method for NPC because the disease is highly sensitive to ionizing radiation¹. Thus, we focused on the effect of USP44 on NPC radiosensitivity.

Clonogenic survival assay is a mainstay of clinical and preclinical radiobiology to test the capability of adherent cells to survive and replicate following insult with radiation²⁻⁴. Thus, we performed clonogenic survival assay to determine the effect of modulation of USP44 expression on NPC cell radiosensitivity. Our results showed that overexpression of USP44 enhanced the radiosensitivity of NPC cells. As the reviewer mentioned, USP44 overexpression can also suppress NPC cell proliferation. Similar to our findings, it has been frequently reported that both

the proliferation and radiosensitivity are affected at the same time⁴⁻⁹.

Actually, in our present study, we used the linear-quadratic (LQ) formula to fit cell survival curves and testify the radiosensitization effect of USP44. The LQ model is based on Chadwick and Leenhouts's theory. LQ formula is expressed as surviving fraction (SF) = $e^{-\alpha D - \beta D^2}$, where α and β are parameters describing the cell's radiosensitivity, and D is the dose to which it is exposed^{3, 10}. Increasing α and/or decreasing β values indicate higher radiosensitivity. SF2 indicates the survival rate after irradiation with 2 Gy, and lower SF2 indicates higher radiosensitivity^{11, 12}. Here, the α value of *USP44* overexpression group significantly increased compared with the vector group (SUNE1 0.30±0.03 vs. 0.41±0.04, $p < 0.05$; HONE1 0.33±0.02 vs. 0.68±0.02, $p < 0.01$), while the change of β value was not significant (SUNE1 0.03±0.01 vs. 0.05±0.02, $p = 0.26$; HONE1 0.05±0.01 vs. 0.03±0.01, $p = 0.19$). The SF2 value of *USP44* overexpression group significantly decreased compared with the vector group (SUNE1 48%±2.2% vs. 36%±2.2%, $p < 0.05$; HONE1 42%±1.2% vs. 23%±0.7%, $p < 0.01$). The sensitivity-enhancement ratio (SER) of SF2 in SUNE1 and HONE1 cells was 1.3 and 1.9, respectively. Obviously, the α values and SER increased, while SF2 values decreased in the *USP44* overexpression group, indicating that overexpression of *USP44* can enhance the radiosensitivity of NPC cells.

References:

1. Chen, Y.-P. et al. Nasopharyngeal carcinoma. *The Lancet* 394, 64-80 (2019).
2. Liu, P.H. et al. An IRAK1-PIN1 signalling axis drives intrinsic tumour resistance to radiation therapy. *Nat Cell Biol* 21, 203-213 (2019).
3. McMahon, S.J. The linear quadratic model: usage, interpretation and challenges. *Phys Med Biol* 64, 01TR01 (2018).
4. Jeong, Y. et al. Role of KEAP1/NRF2 and TP53 Mutations in Lung Squamous Cell Carcinoma Development and Radiation Resistance. *Cancer Discov* 7, 86-101 (2017).
5. Shi, Y. et al. Ibrutinib inactivates BMX-STAT3 in glioma stem cells to impair malignant growth and radioresistance. *Sci Transl Med* 10 (2018).
6. Oweida, A.J. et al. STAT3 Modulation of Regulatory T Cells in Response to Radiation Therapy in Head and Neck Cancer. *J Natl Cancer Inst* 111, 1339-1349 (2019).
7. Candas-Green, D. et al. Dual blockade of CD47 and HER2 eliminates radioresistant breast cancer cells. *Nat Commun* 11, 4591 (2020).
8. Jie, X. et al. USP9X-mediated KDM4C deubiquitination promotes lung cancer radioresistance by epigenetically inducing TGF- β 2 transcription. *Cell Death Differ* 28, 2095-2111 (2021).
9. Osuka, S. et al. N-cadherin upregulation mediates adaptive radioresistance in glioblastoma. *J Clin Invest* 131 (2021).

10. Chadwick, K.H. & L Ee Nhouts, H.P. The Molecular Theory of Radiation Biology. (The Molecular Theory of Radiation Biology, 1981).
11. Jiang, W. et al. 5-Azacytidine enhances the radiosensitivity of CNE2 and SUNE1 cells in vitro and in vivo possibly by altering DNA methylation. PloS one 9, e93273-e93273 (2014).
12. Barendsen, G.W. Parameters of linear-quadratic radiation dose-effect relationships: dependence on LET and mechanisms of reproductive cell death. Int J Radiat Biol 71, 649-655 (1997).

2. In Fig 2c, the authors claim that the combination of IR and USP44 overexpression leads to G2M-phase arrest indicating an essential role of USP44 in DNA damage response. It is not mentioned, however, at what time after irradiation the cell cycle was examined. Examining the cell cycle at different time points after irradiation will give a more accurate representation of possible cell cycle defects associated with persisting DSBs in the USP44 expressing cells.

Response:

Thanks for your insightful suggestions. Actually, we have explored the best time point by harvesting serum-starved SUNE1 cells with or without *USP44* overexpression at 0h, 4h, 8h, 16h, 24h after irradiation for cell cycle analysis (Fig. R1), since IR-induced G2/M-phase cycle arrest has been reported at different time points after irradiation¹³⁻¹⁶. We found that the effect of *USP44* overexpression on the induction of G2/M phase arrest was the most obvious at 8h after irradiation compared with the vector group. Thus, we selected 8h after irradiation for following cell cycle analysis.

In our present study, serum-starved cells were collected 8 h after irradiation to examine the cell cycle. We have mentioned the time point for cell cycle analysis in the Methods part of our manuscript. To address the reviewer’s concerns, we have now added the time points in the Figure legend of our revised manuscript (Page 31, paragraph 1; Page 34, paragraph 1).

Fig. R1 Cell cycle distribution of SUNE1 cells transiently transfected with *USP44* or the empty vector plasmids

were detected at 0h, 4h, 8h, 16h and 24h after IR. The data are presented as the mean \pm SD; * $P < 0.05$, ** $P < 0.01$, *** $P < 0.001$ (two-tailed Student's *t*-test), $n = 3$ independent experiments performed in triplicate.

References:

13. Shafi, A.A. et al. The circadian cryptochrome, CRY1, is a pro-tumorigenic factor that rhythmically modulates DNA repair. *Nat Commun* 12, 401 (2021).
14. Lee, Y.H., Kuo, C.Y., Stark, J.M., Shih, H.M. & Ann, D.K. HP1 promotes tumor suppressor BRCA1 functions during the DNA damage response. *Nucleic Acids Res* 41, 5784-5798 (2013).
15. Dutta, A. et al. Microhomology-mediated end joining is activated in irradiated human cells due to phosphorylation-dependent formation of the XRCC1 repair complex. *Nucleic Acids Res* 45, 2585-2599 (2017).
16. Zumsteg, Z.S. et al. Taselisib (GDC-0032), a Potent π -Sparing Small Molecule Inhibitor of PI3K, Radiosensitizes Head and Neck Squamous Carcinomas Containing Activating PIK3CA Alterations. *Clin Cancer Res* 22, 2009-2019 (2016).

3. *It is not mentioned what conditions were used for the IP/IB experiments used to detect the increased ubiquitination of Ku80 and TRIM25 (Figs 3g and 4d). The authors should perform the experiments shown in these figures in denaturing conditions to confirm that the ubiquitinated species precipitated with FLAG/MYC IPs are polyubiquitinated Ku80 and TRIM25. Blots for ubiquitin (HA) can often detect the increased ubiquitination of associated proteins precipitated with the bait proteins. This concern is supported by the observation that there is a vast difference of the HA-Ub level in the first lanes of the HA blots in Figs 3l and 3m, although these should look similar. At least, the authors can perform these experiments in reverse order, where they precipitate ubiquitinated species and blot for Ku80 or TRIM25 to detect the higher molecular weight, ubiquitinated forms of these proteins in the described conditions.*

Response:

Thanks for your valuable questions. We apologize we did not detailly describe the conditions of the IP/IB experiments in the method parts. Actually, all the ubiquitin assays were performed in denaturing conditions according to the methods used in our previous papers¹⁷⁻¹⁹. In brief, Cells were lysed in regular lysis buffer (100 μ l) and the cell lysates were denatured at 95°C for 5min in the presence of 1% SDS. A portion of cell lysates (20 μ l) were saved for immunoblot analysis to detect the expression of target proteins. The rest of cell lysates (80 μ l) were diluted with 1ml lysis buffer and immunoprecipitated with specific antibodies. The immunoprecipitates were washed by three times and subject to immunoblot analysis (Page 20, paragraph 2; Page 31, paragraph 2; Page 32, paragraph 1-2; Page 33, paragraph 1).

As for the vast difference of the HA-Ub level in the first lanes of the HA blots in Figs 3l and 3m, it is due to the different exposure intensities. Here are the bands at different exposure times of Fig 3l. In order to eliminate the misunderstanding, we have replaced the original strip with a stronger exposure intensity strip (Fig. 3l).

Fig. 3 l, HEK293T cells transfected with the indicated plasmids were subjected to denature-IP and then immunoblotted with the indicated antibody. The HA blots with short exposure (left) and long exposure (right) were shown.

References:

17. Zhao, Y. et al. USP2a Supports Metastasis by Tuning TGF-beta Signaling. *Cell Rep* 22, 2442-2454 (2018).
18. Zhao, Y. et al. Hypermethylation of UCHL1 Promotes Metastasis of Nasopharyngeal Carcinoma by Suppressing Degradation of Cortactin (CTTN). *Cells* 9 (2020).
19. Zheng, Z.Q. et al. Long Noncoding RNA TINCR-Mediated Regulation of Acetyl-CoA Metabolism Promotes Nasopharyngeal Carcinoma Progression and Chemoresistance. *Cancer Res* 80, 5174-5188 (2020).

4. Based on the experiments presented in Fig 4i, the authors concluded that USP44 removes ubiquitin specifically from K439 on TRIM25, and by this, inhibits its degradation. However, there might be several other explanations for the results presented in this figure. First, just like for the WT, the authors should examine the ubiquitination level of the other two mutants (K283/284R and K509R) in cells without USP44 expression, i.e., compare cells lysates with or without USP44 expression for all constructs, not just for the WT. As presented now, it is unknown if the reduced ubiquitination of TRIM25-K283/284R and TRIM25-K509R in the blot results from the USP44 activity or if these mutants cannot be ubiquitinated to the same extent as WT and K439R? Second, it is obvious that multiple lysines in TRIM25 can be ubiquitinated and that the K439R mutant displays almost WT levels of ubiquitination. Thus, it is unknown how much K439

contributes to the total ubiquitination of TRIM25 hence its stability in the cells. Furthermore, K439R may introduce structural changes in TRIM25, which prevent its interaction with USP44 and thus its deubiquitination by this DUB. Therefore, the interaction between TRIM25-K439R and USP44 should be tested.

Response:

Thanks for your valuable comments. Following the reviewer's suggestion, we conducted the denature-IP in SUNE1 cells without USP44 overexpression to examine the ubiquitination levels of all the TRIM25 KR mutants. When without USP44 overexpression, the ubiquitination levels of all TRIM25 mutants (K283/284R, K439R or K509R) were weaker than that of the wild-type TRIM25 (WT), indicating that all TRIM25 KR mutants can be ubiquitinated. With USP44 overexpression, the ubiquitination level of TRIM25 (K439R) was stronger than that of TRIM25 (WT) and TRIM25 (K283/284 and K509R), indicating that the K439R mutant can resist the deubiquitinating of USP44, and the Lys439 contributes to the most ubiquitination of TRIM25 (Fig. 4i). Hence, USP44 recruits TRIM25 and impairs the Lys439-mediated K48-linked ubiquitination of TRIM25 and further inhibits its degradation (Page 9, paragraph 1).

In addition, our truncation co-IP shows that both the HR1 domain of TRIM25 (TRIM25-N2) and the PRY/SPRY domain of TRIM25 (TRIM25-N3) are important for the binding of TRIM25 to USP44 (Supplementary Fig. 5a), so we guess TRIM25 K439R mutant (included in TRIM25-N3) may not affect the binding between TRIM25 and USP44. In order to verify our guess and dispel the doubts of the reviewer, we conducted co-IP experiments by co-transfected HA-tagged USP44 and FLAG-tagged TRIM25 (WT) or TRIM25 (K439R) plasmids in SUNE1 and HONE1 cells. The results showed that the interaction between USP44 and TRIM25 (WT) were the same as that between USP44 and TRIM25 (K439R) (Fig. R2), indicating that the TRIM25 K439R mutation did not affect the structure of TRIM25 and its bind with USP44. Therefore, USP44 did not remove the ubiquitin of TRIM25 (K439R) mutant, not because of the weakening of the interaction between USP44 and TRIM25 (K439R). These results confirmed that USP44 removed ubiquitin specifically from K439 on TRIM25 and inhibited its degradation.

Fig. 4 i, HEK293T cells were transfected with the vector plasmid or HA-USP44, HA-Ub, and Flag-TRIM25 WT or KR mutants, treated with MG132, subjected to denature-IP with anti-Flag beads and then analysed by immunoblotting with an anti-HA or anti-Flag antibody.

Fig. R2 Co-IP with anti-HA antibody in SUNE1 and HONE1 cells transiently transfected with HA-USP44 and FLAG-TRIM25(WT or K439R mutant) revealed the exogenous association of USP44 and TRIM25 (WT or K439R mutant).

5. In Fig 4j, the authors used IF to illustrate that overexpression of USP44 leads to reduced levels of Ku80 in cells and that this effect can be rescued by TRIM25 ablation. The results in this figure, however, are puzzling. It is unclear why IR will induce such a drastic difference in the Ku80 levels in the cells (comparing the Ku80 signal in Ctrl and IR panels). The total levels of Ku80 in cells are usually not affected by IR (which is also supported by the results presented in Fig 5c). As shown, it appears that USP44 affects the Ku80 induction/expression upon irradiation rather than its stability in these cells. Furthermore, the Ku80 signal seems to be the strongest in USP44 overexpressing cells before IR, which contradicts the data presented in the figures before (Figs 3d end 3e).

Response:

Thanks for your valuable advice. It has been reported that IR could induce the increase of Ku80 expression in protein levels²⁰⁻²². To address the reviewer's concerns, we conducted western blot assays to detect the Ku80 expression in Vector+sgNC, USP44+sgNC, and USP44+sgTRIM25 grouped SUNE1 cells with or without IR induction. The results showed that the Ku80 expression was obviously increased upon IR induction. In addition, overexpression of USP44 reduced the Ku80 expression with or without IR, and knockout of TRIM25 could rescue the USP44-mediated Ku80 degradation, which showed the same results as that of immunofluorescence experiments in TRIM25 knocked-down cells (Page 9, Paragraph 1; Supplementary Fig. 7d). In Fig 4j, the whole cells were irradiated by IR in the IR group. As for Fig 5c, live cells were micro-irradiated with a UVA laser, as red dotted lines indicate, and DSBs formed only on the line micro-irradiated with a UVA laser. Thus, there is no comparability between Fig 4j and Fig 5c. To address the reviewer's concerns, we repeated the immunofluorescence experiment and counted the mean intensity of Ku80 fluorescence. We found that the Ku80 signal was the weakest in USP44 overexpressing cells before IR, although the fluorescence intensity before IR is very weak in all of the three groups. These results demonstrated that USP44 decreased the stability of Ku80 through TRIM25 in NPC cells. (Page 9, Paragraph 1; Fig. 4j; Supplementary Fig. 7b).

Fig. 4 j, SUNE1 and HONE1 cells transfected with the indicated plasmids and siRNAs were exposed to IR (6 Gy), fixed 0.5 h later and co-immunostained with the anti-Ku80 antibody. Scale bars, 10 μ m.

Supplementary Fig. 7 b, The mean intensity of Ku80 fluorescence in fig. 4j was quantified. **d**, Western blot analysis of the Vector+sgNC, USP44+sgNC, and USP44+sgTRIM25 grouped SUNE1 cells with IR treatment (6 Gy, 0.5 h) or not. Data in b are presented as the mean \pm SD, *** P < 0.001 (two-tailed Student's t -test), n = 20 independent experiments performed in triplicate.

References:

- He, Y. et al. Long non-coding RNA PVT1 predicts poor prognosis and induces radioresistance by regulating DNA repair and cell apoptosis in nasopharyngeal carcinoma. *Cell Death & Disease* 9, 235 (2018).
- Zhang, Q. et al. FBXW7 Facilitates Nonhomologous End-Joining via K63-Linked Polyubiquitylation of XRCC4. *Molecular Cell* 61, 419-433 (2016).
- Sharma, A. et al. USP14 is a deubiquitinase for Ku70 and critical determinant of non-homologous end joining repair in autophagy and PTEN-deficient cells. *Nucleic Acids Res* 48, 736-747 (2020).

6. In Fig 5, the authors present evidence for DNA repair defects in cells overexpressing USP44. However, the data shown in Fig 5a and Fig 5b suggest that the USP44 overexpressing cells are more susceptible to DSBs upon IR rather than have DSBs repair defects. There is no explanation why USP44 expressing cells will form approximately four times longer comet tails 30 min after the same dose of irradiation? It looks like that most of the damage is then repaired in 24 h. The same is true for the γ H2AX foci experiment (Fig 5b), where the USP44 overexpressing cells form more foci than vector-only cells. There is no indication at what time after the irradiation the cells were stained for γ H2AX. To claim that USP44 overexpression impairs DSBs repair, the authors should perform a time kinetic where they compare the changes in the tail moment or foci number over time, starting with comparable amount foci/tail lengths in all cell lines.

Response:

Thanks for your valuable comments. It has been reported that the tail moments and γ H2AX foci could be obviously induced at 0.5h after IR treatment²³. As the reviewer suggested, we performed a time kinetic where they compare the changes in the tail moment or foci number at 0h, 0.5h, 4h and 24h after IR treatment in Vector+sgNC, USP44+sgNC, and USP44+sgTRIM25

grouped SUNE1 or HONE1 cells (Supplementary Fig. 8a). Our results showed that while the level of DNA damage indicated by comet tail gradually returned to baseline in the Vector+sgNC cells 24 h after IR treatment, it remained higher in the USP44+sgNC cells, suggesting there were delays in DNA repair in the USP44 overexpression cells. Moreover, TRIM25 knockout reversed these DNA damage, suggesting that USP44 has a negative impact on DSB repair by targeting TRIM25 (Fig. 5a). This was further confirmed by differences in the levels of γ H2AX foci induced by IR at different time points. Overexpression of USP44 enhanced the formation of DSB marker γ H2AX foci induced by IR, which could be reversed by knockout of TRIM25 (Fig. 5b, top). Consistently, the number of γ H2AX-positive foci quickly diminished in the Vector+sgNC cells but was sustained in the USP44+sgNC cells, which were reversed in the USP44+sgTRIM25 cells (Fig. 5b, bottom). These results indicate that DSB repair activity is impaired by USP44 overexpression, which could be reversed by TRIM25 knockout (Page 9, paragraph 2; Page 10, Paragraph 1; Fig. 5a, b; Supplementary Fig. 8a).

Supplementary Fig. 8 a, Western blot analysis of the Vector+sgNC, *USP44*+sgNC, and *USP44*+sg*TRIM25* grouped SUNE1 and HONE1 cells.

Fig. 5 a, Representative comet images (top) and quantitative analysis of tail moments (bottom) for 6Gy-IR induced DNA damage in the indicated NPC cells, measured by the comet assay. Scale bars, 10 μ m. **b**, Representative images (top) and quantitative analysis (bottom) of the number of γ H2AX foci in the indicated NPC cells with or without 6Gy-IR exposure. Scale bars, 10 μ m. Data in a and b are presented as the mean \pm SD, *** P < 0.001 (two-tailed Student's t -test), n = 20 (b), n = 10 (c) independent experiments performed in triplicate.

References:

23. Zhang, Y. et al. Long noncoding RNA LINP1 regulates repair of DNA double-strand breaks in triple-negative breast cancer. *Nat Struct Mol Biol* 23, 522-530 (2016).

7. The data in Fig 5c will argue that the USP44 functions are not as crucial for the total Ku80 levels in the cells but rather for loading on DSBs upon damage. The levels of GFP-Ku80 in HONE1 cells are not significantly different in USP44 overexpressing cells compared to the Vector only and USP44+TRIM25KO cells, yet they failed to form laser-induced Ku80 foci/streaks.

Response:

Thanks for your valuable comments. As is well known, fluorescent signals could be bleached by long-time exposure to fluorescent light sources. Consecutive images were captured at 20s interval for 10min with a Nikon Elipse TI2-U inverted microscope. We manually turned off the fluorescent light source during the capture interval. However, the manual operation was not agile enough, so that the cells were still exposed to fluorescent light source for at least 5s before and

after each capture, which led to the bleaching of fluorescence signals. Thus, the total Ku80 levels in Fig 5c were unstable in the same cell.

To address the reviewer's concerns, we repeated the laser microirradiation assay for live-cell imaging with a Nikon AX confocal microscope, which could be automatically set to turn on the fluorescent light source only at each capture to avoid the bleaching of fluorescence signals. Consecutive images were captured at 10s interval for 10min. Our results showed that the levels of GFP-Ku80 in both SUNE1 and HONE1 cells are significantly lower in *USP44* overexpressing cells compared to the Vector+sgNC and *USP44*+sg*TRIM25* cells. These cells were micro-irradiated with a 65% energy UVA laser indicated by red dotted lines, but *USP44* overexpression remarkably reduced the intensity of laser-induced Ku80 streaks. Our results showed that *USP44* overexpression resulted in impaired either of the total Ku80 levels in the cells or the recruitment of GFP-Ku80 at DSB sites, and this effect could be largely reverted by *TRIM25* knockout (Page 10, Paragraph 1; Page 13, Paragraph 2; Page 20, Paragraph 4; Fig. 5c).

Fig. 5 c, The indicated NPC cells were transfected with GFP-Ku80 and then subjected to laser micro-IR and live-cell imaging. Scale bars: 10 μ m.

8. It will be important to see if higher levels of *USP44* correlate with low levels of *Ku80* in patients (Fig 8a).

Response:

Thanks for your valuable advice. As the reviewer suggested, we did immunohistochemical staining of 20 NPC tissues with antibodies against anti-*USP44* or anti-*Ku80*. Our results indicated that higher levels of *USP44* correlate with lower levels of *Ku80* in NPC tissue samples (Page 12, Paragraph 1; Supplementary Fig. 9a, b).

Supplementary Fig. 9 a, Representative images of immunohistochemical staining anti-USP44 or anti-Ku80 in NPC patients. Scale bars, 50 μ m. **b**, Correlation analysis of USP44 expression and Ku80 expression in NPC samples ($n = 20$) according to IHC score statistics.

Minor concerns:

1. There is a discrepancy in the text and the label of Fig 5a. The text mentions that the tail moment remains higher in USP44 overexpressing cells for four hours, whereas the last point in the figure is 24h.

Response:

Thank you for point out this mistake. The tail moment remains higher in USP44 overexpressing cells for 24 hours and we have corrected the relevant part in our revised manuscript (Page 9, Paragraph 2, Fig. 5a).

2. There is no evidence that USP44 binds DNA/chromatin (directly); it will be more accurate if the authors depict it not as a DNA-bound protein in their model figure.

Response:

Thank you for your valuable suggestion. As the reviewer said, we did not prove that USP44 directly bond to DNA or chromatin. Our research demonstrated that USP44 recruited TRIM25 to degrade Ku80 upon IR induction, indicating that USP44 was recruited to DNA double-strand breaks via interacting with TRIM25 and Ku80. For the sake of the preciseness of the conclusion of the article, we followed the reviewer's suggestion and modified the model figure (Fig. 8i).

i

Fig. 8i, Proposed working model of USP44. USP44 recruits and stabilizes TRIM25 by removing the K48-linked polyubiquitin chains of TRIM25, and TRIM25 degrades Ku80 by promoting its polyubiquitination and inhibits its recruitment to DSBs, which further inhibits the NHEJ pathway and enhances NPC radiosensitivity. In NPC, hypermethylation of the USP44 promoter leads to its downregulation at the mRNA and protein levels, which blocks the anticancer effect of the USP44-TRIM25-Ku80 axis.

Reviewer #2 (Remarks to the Author):

Summary:

Chen et al present data in nasopharyngeal carcinoma (NPC) that expression of the protein USP44 regulates the sensitivity of NPC to radiotherapy. Through analysis of a previously published data set they show that USP44 promoter is hypermethylated in NPC relative to normal epithelium and these results are corroborated in pre-clinical cell line models. They then show that overexpression of USP44 results in increased radiosensitivity and conversely that depletion of USP44 via RNAi leads to relative radioresistance, reportedly through modulation of apoptosis, cell cycle arrest and changes in DNA damage levels. The mechanism is reported to involve the interaction between USP44, TRIM25 and Ku80 (a known regulator of the DNA damage response/NHEJ). Finally, their results are demonstrated in an in vivo animal model and with analysis.

Comments:

1. Overall the data are interesting and are in line with prior results suggesting that USP44 is involved in DNA Repair and recruitment of DNA repair factors such as 53BP1 to sites of DSBs (Ref 23). One significant weak point of this manuscript is the translational/therapeutic relevance of these findings. The authors do not report upon (or even discuss) methods by which these findings may be exploited for therapeutic gain in patients with NPC.

Response:

Thank you for your valuable comments. In our research, we show that hypermethylation of *USP44* promotes radiotherapy resistance in NPC. *USP44* is hypermethylated in NPC, which is associated with its downregulation in NPC and many other types of tumours. *USP44* enhances the sensitivity of NPC cells to radiotherapy *in vitro* and *in vivo* through the USP44-TRIM25-Ku80 axis. USP44 recruits and stabilizes TRIM25 by removing its K48-linked polyubiquitin chains at Lys439, which further facilitates the degradation of Ku80 and inhibits its recruitment to DSBs, thus enhancing DNA damage and inhibiting NHEJ-mediated DNA repair. Low expression of USP44 is associated with tumour relapse and a poor prognosis in NPC patients. USP44 plays a role as a tumor suppressor gene as usually, like a safeguard as p53 and Rb in the normal tissues. P53 can sense DNA damage, caused by radiation or drugs, and these cells then initiated the DNA damage repair response and then enter the normal proliferation cycle, or enter the death pathway

such as apoptosis, necrosis and so on, so as to prevent the abnormal cells from entering the cell cycle to develop into malignant cells^{1, 2}. The Rb protein can specifically bind to transcriptional cofactor E2F and inactivate it, which further inhibits the expression of genes necessary for the transition from G1 phase to S phase. As a tumor suppressor, Rb can also play an important role in the maintenance of genome stability and apoptosis³⁻⁶. Besides, there are multiple preclinical studies which are focused on the mechanistic analysis of tumor suppressor genes on tumor radiosensitivity⁷⁻¹⁰. Similar studies can enrich the theories of tumorigenesis and broaden the understanding of the mechanisms of radiotherapy resistance.

DNA methylation, especially the transcriptional inactivation of tumor suppressors caused by CpG island methylation, is closely related to the process of tumorigenesis and development^{11, 12}. Demethylation drugs can reverse methylation and restore the expression of tumor suppressor genes for the treatment of tumors^{13, 14}. For example, 5-azacytidine and 5-aza-2'-deoxycytidine were approved by FDA for the treatment of myelodysplastic syndrome. At present, there is no method to specifically remove the methylation of USP44 and restore its expression, but perhaps advanced technology can make it in the future. Besides, the research on the mechanisms of ubiquitination and related drugs is developing rapidly. For example, proteolysis-targeting chimeras (PROTACs) hijacked a E3 ubiquitin ligase to form a ternary complex with a target protein and promoted its polyubiquitination and degradation of the target protein, which made it possible to target many proteins that were previously considered undruggable¹⁵⁻¹⁷. In our research of USP44-TRIM25-KU80 signal axis, TRIM25 as a E3 ubiquitin ligase can be hijacked to target Ku80 through PROTACs to develop a specific treatment for NPC. Therefore, our research has laid a foundation for better understanding the mechanisms of radioresistance and the development of new treatments for NPC. To address the reviewer's concerns, we have added these points in the Discussion Part in our revised manuscript (Page 14, paragraph 2).

References:

1. Kannan, K. et al. DNA microarray analysis of genes involved in p53 mediated apoptosis: activation of Apaf-1. *Oncogene* 20, 3449-3455 (2001).
2. Mirza, A. et al. Global transcriptional program of p53 target genes during the process of apoptosis and cell cycle progression. *Oncogene* 22, 3645-3654 (2003).
3. Classon, M. & Harlow, E. The retinoblastoma tumour suppressor in development and cancer. *Nat. Rev. Cancer* 2, 910-917 (2002).
4. Hernando, E. et al. Rb inactivation promotes genomic instability by uncoupling cell cycle

- progression from mitotic control. *Nature* 430, 797-802 (2004).
5. Zhang, H.S., Postigo, A.A. & Dean, D.C. Active transcriptional repression by the Rb-E2F complex mediates G1 arrest triggered by p16INK4a, TGFbeta, and contact inhibition. *Cell* 97, 53-61 (1999).
 6. Hallstrom, T.C., Mori, S. & Nevins, J.R. An E2F1-dependent gene expression program that determines the balance between proliferation and cell death. *Cancer Cell* 13, 11-22 (2008).
 7. Chen, Y. et al. LRR31 inhibits DNA repair and sensitizes breast cancer brain metastasis to radiation therapy. *Nat. Cell Biol.* 22, 1276-1285 (2020).
 8. Binkley, M.S. et al. KEAP1/NFE2L2 Mutations Predict Lung Cancer Radiation Resistance That Can Be Targeted by Glutaminase Inhibition. *Cancer Discov.* 10, 1826-1841 (2020).
 9. Yun, E.J. et al. Downregulation of Human DAB2IP Gene Expression in Renal Cell Carcinoma Results in Resistance to Ionizing Radiation. *Clin Cancer Res* 25, 4542-4551 (2019).
 10. Wang, F. et al. SMAD4 Gene Mutation Renders Pancreatic Cancer Resistance to Radiotherapy through Promotion of Autophagy. *Clin Cancer Res* 24, 3176-3185 (2018).
 11. Flavahan, W.A., Gaskell, E. & Bernstein, B.E. Epigenetic plasticity and the hallmarks of cancer. *Science* 357, eaal2380 (2017).
 12. Koch, A. et al. Analysis of DNA methylation in cancer: location revisited. *Nat. Rev. Clin. Oncol.* 15, 459-466 (2018).
 13. Edlin, R. et al. Azacitidine for the treatment of myelodysplastic syndrome, chronic myelomonocytic leukaemia and acute myeloid leukaemia. *Health Technol. Assess.* 14 Suppl 1, 69-74 (2010).
 14. Steensma, D.P. et al. Multicenter study of decitabine administered daily for 5 days every 4 weeks to adults with myelodysplastic syndromes: the alternative dosing for outpatient treatment (ADOPT) trial. *J. Clin. Oncol.* 27, 3842-3848 (2009).
 15. Dale, B. et al. Advancing targeted protein degradation for cancer therapy. *Nat. Rev. Cancer* (2021).
 16. Bondeson, D.P. et al. Catalytic in vivo protein knockdown by small-molecule PROTACs. *Nat. Chem. Biol.* 11, 611-617 (2015).
 17. Winter, G.E. et al. DRUG DEVELOPMENT. Phthalimide conjugation as a strategy for in vivo target protein degradation. *Science* 348, 1376-1381 (2015).

2. Along these lines, generally there is the expectation that a target should be tumor specific (so as to not increase normal tissue toxicity), and the authors have not shown that modulation of USP44 in normal tissues would not lead to increased sensitivity to radiotherapy or other DNA damaging agents.

Response:

Thank you for your valuable comments. Actually, we do want to conduct clonogenic assays in normal nasopharyngeal epithelial cells NP69 or N2-Tert transfected with shCtrl or shUSP44 plasmids upon IR to examine its effect on the sensitivity to radiotherapy. However, NP69 or N2-Tert cells cannot form colonies after IR, owing to that they do not have the properties as tumor

cells possess, such as the ability of infinite proliferation and stemness.

According to our working model, in normal tissues, USP44 recruits and stabilizes TRIM25 by removing the K48-linked polyubiquitin chains of TRIM25, and then TRIM25 degrades Ku80 by promoting the polyubiquitination of Ku80, which inhibits its recruitment to DSBs and further impairs NHEJ-mediated DNA repair and enhances NPC radiosensitivity. In NPC, hypermethylation of the USP44 promoter leads to its downregulation at the mRNA and protein levels, which blocks the anti-tumour effect of the USP44-TRIM25-Ku80 axis. USP44 exerts an effect of tumor suppressor in the normal tissues. The mutation, deletion or inactivation of tumor suppressor genes can lead to malignant transformation of cells and tumorigenesis¹⁸⁻²⁰. Many tumor suppressor genes have been reported to regulate the cell cycle, apoptosis and other process to influence tumor radiosensitivity as described in the response of comment 1. These studies help us to better understand the molecular regulation network of tumor radioresistance. Because DNA methylation is a reversible epigenetic modification process, demethylation drugs can reverse methylation and restore the expression of tumor suppressor genes for the treatment of tumors^{21,22}. At present, there is no method to specifically remove the methylation of USP44 and restore its expression, but perhaps advanced technology can make it in the future. Besides, we explored the mechanism of USP44-TRIM25-KU80 signal axis and its effect on NPC radiosensitivity. PROTACs hijacked a E3 ubiquitin ligase to degrade the target protein, which made it possible to specifically target Ku80 by TRIM25 as described in the response of comment 1. Thus, our research has broadened the understanding of the mechanisms of radioresistance and provided a new strategy for the development of NPC treatment methods.

References:

18. Sherr, C.J. & McCormick, F. The RB and p53 pathways in cancer. *Cancer Cell* 2, 103-112 (2002).
19. Stiewe, T. The p53 family in differentiation and tumorigenesis. *Nat. Rev. Cancer* 7, 165-168 (2007).
20. van Deursen, J.M. Rb loss causes cancer by driving mitosis mad. *Cancer Cell* 11, 1-3 (2007).
21. Edlin, R. et al. Azacitidine for the treatment of myelodysplastic syndrome, chronic myelomonocytic leukaemia and acute myeloid leukaemia. *Health Technol. Assess.* 14 Suppl 1, 69-74 (2010).
22. Steensma, D.P. et al. Multicenter study of decitabine administered daily for 5 days every 4 weeks to adults with myelodysplastic syndromes: the alternative dosing for outpatient treatment (ADOPT) trial. *J. Clin. Oncol.* 27, 3842-3848 (2009).

3. While the authors have utilized overexpression experiments to show that USP44 is responsible for modulating radiation sensitivity, they have utilized RNAi based techniques to deplete USP44 expression. RNAi techniques are well known to have substantial off-target effects with regards to DNA repair (PMID 22344029) processes and as such use of CRISPR based techniques should be done to confirm that depletion of USP44 modulates radiation survival. In these same experiments the western blots in Suppl Figure 2b do not show convincing changes in USP44 expression between shControl and shUSP44 expression. Furthermore, expression in the WT, USP44 knockdown cells and USP44 overexpressed cells should be compared on the same western blot to understand the differences in expression required for changes in radiation response.

Response:

Thanks for your valuable suggestions. Following the reviewer's suggestion, we conducted the clonogenic assays in SUNE1 wild type (WT) and USP44 knockout (KO) cells produced by sgUSP44 transfection. The results showed that knockout of *USP44* in SUNE1 cells significantly promoted cell survival after DNA damage caused by IR (Supplementary Fig. 2f), which were the same as that in USP44 knocked-down NPC cells by shRNAs. The above experiments indicated that the shRNAs targeted USP44 were specific and the conclusion that USP44 improved cell survival after DNA damage caused by IR was solid (Page 5, Paragraph 2).

Besides, we conducted the western blot assays to detect the expression of USP44 in the WT, USP44 knocked-down, and USP44 overexpressed cells. The results showed that the expression of USP44 in the *USP44* stably overexpression group were much higher than that in the Vector group, and the expression of USP44 in the USP44 knocked-down group were much lower than that in the shcon group (Supplementary Fig. 2a, b). The shUSP44 #1/2 plasmids were transiently transfected into SUNE1 or HONE1 cells with stable USP44 overexpression (because of the low expression of USP44 in NPC cells). Thus, there is no comparability of USP44 expression between WT NPC cells and shcon NPC cells. We apologize for not describing clearly and now we have added the detailed description in the method parts in our revised manuscript (Page 16, Paragraph 3).

Supplementary Fig. 2 a, b, Western blot analysis of SUNE1 and HONE1 cells with *USP44* overexpression (a) or knockdown (b). The empty vector or HA-*USP44* plasmids were stably transfected into SUNE1 or HONE1 cells.

The control or sh*USP44* plasmids (sh1 or sh2) were transiently transfected into SUNE1 or HONE1 cells with stable *USP44* overexpression. **f**, Clonogenic assays (f, left), survival fraction curves (f, right) after exposure to indicated IR of sgNC or sg*USP44* SUNE1 cells. The sgNC or sg*USP44* cells were constructed upon SUNE1 cells with stable *USP44* overexpression. Data in f are presented as the mean ± SD, ****P* < 0.001 (two-tailed Student's *t*-test), *n* = 3 independent experiments performed in triplicate.

4. Differences attributed to survival we thought to be due to changes in *Ku80* driven by *USP44* and *TRIM25* interaction, however no experiments were done to determine whether this is true (i.e. through re-expression of *Ku80* in *USP44* overexpressing cells).

Response:

Thanks for your valuable suggestions. We have performed clonogenic assays to see the effects of knockdown of *TRIM25* in *USP44* overexpressing cells on radiation survival in Fig. 6a. Following the reviewer's suggestion, we conducted the clonogenic assays in SUNE1 and HONE1 cells transfected with the empty vector or *USP44* plus the empty vector or *Ku80* plasmids to see the effects of re-expression of *Ku80* in *USP44* overexpressing cells on radiation survival. The results showed that overexpression of *USP44* in SUNE1 or HONE1 cells significantly inhibited cell survival after DNA damage caused by IR, which could be reversed by re-expression of *Ku80*. These results indicated that *Ku80* decrease driven by *USP44* and *TRIM25* interaction contributed to the suppressive effects on NPC cell survival (Page 10, Paragraph 2; Supplementary Fig. 8e).

Supplementary Fig. 8 e, Clonogenic assays (top), survival fraction curves (bottom) after exposure to indicated IR of SUNE1 and HONE1 cells transiently co-transfected with HA-USP44 or the empty vector plus FLAG-TRIM25 or the empty vector plasmids. Data are presented as the mean \pm SD, ** $P < 0.01$, *** $P < 0.001$ (two-tailed Student's t -test), $n = 3$ independent experiments performed in triplicate.

5. There are no statistical tests done to compare cell cycle profiles seen in several figures.

Response:

Thank you for your kind reminding. Data of cell cycle distribution are presented as the mean of $n = 3$ independent experiments. It is our negligence that no statistical tests were done to compare cell cycle profiles seen in several figures. We have added statistics to the corresponding graphs and these statistical charts was presented as the mean \pm SD in our revised manuscript (Fig. 2c; Fig. 6c; Supplementary Fig. 3a).

Fig. 2 c, Cell cycle distribution of SUNE1 and HONE1 cells transiently transfected with USP44 or the empty vector plasmids with or without exposure to 6-Gy IR. Cell cycle distribution was detected at 8h after IR and apoptosis rate was detected at 24h after IR. Data are presented as the mean \pm SD; *** $P < 0.001$ (two-tailed Student's t -test), $n = 3$ independent experiments performed in triplicate.

Fig. 6 c, Cell cycle distribution of SUNE1 and HONE1 cells transiently co-transfected with HA-USP44 or the empty vector plasmids plus siTRIM25 or control siRNA with or without exposure to 6-Gy IR. Cell cycle distribution was detected at 8h after IR and apoptosis rate was detected at 24h after IR. Data are presented as the mean \pm SD; * $P < 0.05$, *** $P < 0.001$ (two-tailed Student's t -test), $n = 3$ independent experiments performed in triplicate.

Supplementary Fig. 3 a, Cell cycle distribution of SUNE1 and HONE1 cells with or without exposure to 6-Gy IR. The control or shUSP44 plasmids (sh1 or sh2) were transiently transfected into SUNE1 or HONE1 cells with stable USP44 overexpression. Cell cycle distribution was detected at 8h after IR and apoptosis rate was detected at 24h after IR. Data are presented as the mean \pm SD; * $P < 0.05$, ** $P < 0.01$, *** $P < 0.001$ (two-tailed Student's t -test), $n = 3$ independent experiments performed in triplicate.

6. Given the known role of USP44 and regulation of mitosis it would be interesting to understand whether the G2/M checkpoint was altered (e.g., using phosphor Histone H3 staining to quantify mitotic cells) in their experimental models. Additionally, the authors should attempt to discuss/connect the impact of USP44 expression leading to changes in Ku80 expression, NHEJ efficiency and the results seen in Figure 2 (apoptosis and cell cycle arrest).

Response:

Thanks for your insightful suggestions. Following your suggestion, the immunofluorescence

experiments against H3S10_p (a marker of G2/M checkpoint) and γ H2AX (a marker of DSBs) were conducted to see whether the G2/M checkpoint was altered in sgNC or sgUSP44 SUNE1 cells with or without IR induction upon nocodazole treatment. The results showed that the γ H2AX fluorescence was enhanced upon IR induction, and the H3S10_p fluorescence was also enhanced upon IR induction, which means that IR treatment can promote G2/M cell cycle arrest in SUNE1 cells. Besides, we found that the H3S10_p fluorescence were stronger in sgNC group than that in sgUSP44 group upon IR treatment, indicating that knockout of USP44 inhibited the IR-induced G2/M cell cycle arrest, which was consistent with the previous observations in the apoptosis experiments (Page 6, paragraph 1; Supplementary Fig. 3c).

USP44 has been reported to act as a tumour suppressor that regulates cell cycle arrest and DSB responses by modulating H2B mono-ubiquitylation^{23, 24}. Our study showed that USP44 arrested NPC cells in G2/M phase indicated by H3S10_p fluorescence. USP44 could also cause G2/M phase arrest by preventing the premature activation of APC to regulate mitotic checkpoint and binding to the centriole protein centrin to regulate centrosome positioning^{24, 25}. We found that USP44 promoted G2/M phase arrest, apoptosis induction and radiosensitization of NPC through the TRIM25-Ku80 axis *in vivo* and *in vitro*. One of the most common effects of IR is cell cycle arrest²⁶. An increasing proportion of cells in G2/M phase indicates that cells are more sensitive to IR²⁷⁻²⁹. DNA damage after IR also leads to a strong cell apoptosis response³⁰. DNA double-strand breaks (DSBs) are the most critical type of DNA damage induced by IR, and the majority of DSBs are repaired via non-homologous end joining (NHEJ) pathway^{31, 32}. The Ku80-Ku70 heterodimer binds rapidly and tightly to the ends of DSBs and further recruits many other factors required for NHEJ-mediated DNA repair, including DNA-dependent protein kinase catalytic subunit (DNA-PKcs), the XRCC4-LIG4-XLF ligation complex, and APTX and APTF proteins; thus, Ku80 plays an essential role in the initiation of the NHEJ-mediated DNA repair pathway^{33, 34}. These responses induced by IR are independent and interrelated, and all of them can affect the radiosensitivity of tumor cells. USP44-TRIM25 could degrade Ku80 to inhibit NHEJ-mediated DNA repair, which combined with G2/M phase arrest and apoptosis induction to subsequently enhance radiosensitivity in NPC. USP44 impaired the recruitment of Ku80 at DSBs upon laser micro-IR, and this effect could be largely rescued by TRIM25 depletion. This finding reveals a new mechanism by which USP44 regulates DSB repair and radiotherapy resistance by targeting

TRIM25-Ku80 axis for ubiquitination (Page 13, paragraph 2).

Supplementary Fig. 3 c, The immunofluorescence analysis (anti-H3S10p or anti- γ H2AX) in sgNC or sgUSP44 SUNE1 cells treated with nocodazole (0.1 μ g/ml) for 18 hours before IR treatment (6 Gy, 0.5 hour) or not. Scale bar, 10 μ m.

References:

23. Mosbech, A., Lukas, C., Bekker-Jensen, S. & Mailand, N. The Deubiquitylating Enzyme USP44 Counteracts the DNA Double-strand Break Response Mediated by the RNF8 and RNF168 Ubiquitin Ligases. *J. Biol. Chem.* 288, 16579-16587 (2013).
24. Stegmeier, F. et al. Anaphase initiation is regulated by antagonistic ubiquitination and deubiquitination activities. *Nature* 446, 876-881 (2007).
25. Zhang, Y. et al. USP44 regulates centrosome positioning to prevent aneuploidy and suppress tumorigenesis. *J. Clin. Invest.* 122, 4362-4374 (2012).
26. Patties, I., Jahns, J., Hildebrandt, G., Kortmann, R.-D. & Glasow, A. Additive Effects of 5-Aza-2'-deoxycytidine and Irradiation on Clonogenic Survival of Human Medulloblastoma Cell Lines. *Strahlenther. Onkol.* 185, 331-338 (2009).
27. Hao, S. et al. Protein phosphatase 2A inhibition enhances radiation sensitivity and reduces tumor growth in chordoma. *Neuro Oncol.* 20, 799-809 (2018).
28. Wang, Y., Mei, H., Shao, Q., Wang, J. & Lin, Z. Association of ribosomal protein S6 kinase 1 with cellular radiosensitivity of non-small lung cancer. *Int J Radiat Biol* 93, 581-589 (2017).
29. Ho, W.S. et al. LB-100, a novel Protein Phosphatase 2A (PP2A) inhibitor, sensitizes malignant meningioma cells to the therapeutic effects of radiation. *Cancer Lett.* 415, 217-226 (2018).
30. Gulen, M.F. et al. Signalling strength determines proapoptotic functions of STING. *Nat. Commun.* 8, 427-427 (2017).
31. Ciccia, A. & Elledge, S.J. The DNA damage response: making it safe to play with knives. *Mol. Cell* 40, 179-204 (2010).

32. Price, B.D. & D Andrea, A.D. Chromatin Remodeling at DNA Double-Strand Breaks. *Cell* 152, 1344-1354 (2013).
33. Symington, L.S. & Gautier, J. Double-Strand Break End Resection and Repair Pathway Choice. *Annu. Rev. Genet.* 45, 247-271 (2011).
34. Lieber, M.R. The Mechanism of Double-Strand DNA Break Repair by the Nonhomologous DNA End-Joining Pathway. *Annual Review of Biochemistry* 79, 181-211 (2010).

7. It is unclear as to the functional significance/magnitude of change of the NHEJ assay changes seen in Figure 5d. This would be clarified by the addition of proper controls to this assay (e.g., including a knockdown of a known essential NHEJ protein).

Response:

Thanks for your valuable suggestions. For NHEJ reporter assay, when the EJ5-GFP plasmids are transfected into NPC cells, GFP will not be produced. While if we infect the cells with the adenoviruses expressing endonuclease I-SceI, the endonuclease I-SceI will recognize and cut the I-SceI sites to produce DSBs, then if the DSBs are repaired through NHEJ-mediated pathway, the GFP will be restored in NPC cells. Thus, the higher GFP expression indicated higher efficiency of NHEJ repair³⁵⁻³⁷ (Page 10, paragraph 2; Page 21, paragraph 2; Supplementary Fig. 8b). Following the reviewer's suggestion, we repeated the NHEJ repair assay and added a knockdown of a known essential NHEJ protein *Ku70* to verify the functional significance of the NHEJ reporter assay. Our results showed that knockdown of *Ku70* significantly reduced the GFP expression with I-SceI adenoviruses infection (Page 10, paragraph 2; Supplementary Fig. 8c, d). Overexpression of *USP44* also significantly decreased the GFP expression, and knockout of *TRIM25* significantly restored the GFP expression that reduced by *USP44* overexpression. These results demonstrated that *USP44* significantly inhibited NHEJ-mediated DNA repair by targeting *TRIM25* in NPC cells (Page 10, paragraph 2; Fig. 5d; Supplementary Fig. 8a).

Fig. 5 d, The Vector+sgNC, *USP44*+sgNC, and *USP44*+sg*TRIM25* grouped SUNE1 cells were transfected with EJ5-GFP, infected with or without I-SceI adenovirus and analysed for GFP positivity by flow cytometry. Data are presented as the mean \pm SD, ****P* < 0.001 (two-tailed Student's *t*-test), *n* = 3 independent experiments. The data shown are representative of three independent experiments.

Supplementary Fig. 8 a, Western blot analysis of the Vector+sgNC, *USP44*+sgNC, and *USP44*+sg*TRIM25* grouped SUNE1 and HONE1 cells. **b**, Schematic of the EJ5-GFP reporter used to monitor NHEJ repair in NPC cells (see text for details). **c**, Western blot analysis of SUNE1 and HONE1 cells transfected with siNC or *siKu70*. **d**, The siNC or *siKu70* SUNE1 cells were transfected with EJ5-GFP, infected with or without I-SceI adenovirus and analysed for GFP positivity by flow cytometry. Data in d are presented as the mean \pm SD, *** P < 0.001 (two-tailed Student's *t*-test), n = 3 independent experiments. The data shown are representative of three independent experiments.

References:

35. Zhang, Q. et al. FBXW7 Facilitates Nonhomologous End-Joining via K63-Linked Polyubiquitylation of XRCC4. *Mol. Cell* 61, 419-433 (2016).
36. Bennardo, N., Cheng, A., Huang, N. & Stark, J.M. Alternative-NHEJ is a mechanistically distinct pathway of mammalian chromosome break repair. *PLoS Genet.* 4, e1000110 (2008).
37. Ishida, N. et al. Ubiquitylation of Ku80 by RNF126 Promotes Completion of Nonhomologous End Joining-Mediated DNA Repair. *Molecular and Cellular Biology* 37, e00347-00316 (2017).

8. There should be quantitation of the changes in Figure 4J as well as western blot analysis of *Ku80* expression changes with these experimental manipulations.

Response:

Thanks for your useful suggestions. We have added statistics to the corresponding graphs to quantitative the changes of *Ku80* (Supplementary Fig. 7b), and we also conducted the western blot assays to detect *Ku80* expression in the Vector+sgNC, *USP44*+sgNC, and *USP44*+sg*TRIM25* grouped SUNE1 cells with or without IR. The results showed that knockout of *TRIM25* could rescue the *USP44*-mediated *Ku80* degradation after IR induction, which showed the same results as that of immunofluorescence experiments in *TRIM25* knocked-down cells (Page 9, Paragraph 1; Supplementary Fig. 7d).

Supplementary Fig. 7 b, The mean intensity of Ku80 fluorescence in fig. 4j was quantified. **d**, Western blot analysis of the Vector+sgNC, *USP44*+sgNC, and *USP44*+sg*TRIM25* grouped SUNE1 cells with IR treatment (6 Gy, 0.5 h) or not. Data in b are presented as the mean \pm SD, *** $P < 0.001$ (two-tailed Student's *t*-test), $n = 20$ independent experiments performed in triplicate.

9. The radiation dose and timepoints should be specified explicitly throughout the paper. For instance, I do not see this specified for figure 5b.

Response:

Thanks for your kind reminding. It was our negligence that the radiation dose and timepoints were not specified explicitly. We have carefully checked and added the dose and timepoints of radiation experiments in related figure legend parts throughout the paper (Page 31, paragraph 1-2; Page 33, paragraph 2; Page 34, paragraph 1).

Reviewer #3 (Remarks to the Author):

To increase the efficacy of radiotherapy in the treatment of nasopharyngeal cancer, it is essential to understand the mechanism underlying resistance to radiation in cancer. In this manuscript, the authors show that the novel USP44-TRIM25-Ku80 axis plays a critical role in radiation resistance frequently observed in nasopharyngeal cancer. In general, the experiments are well designed, and the results are properly presented. If the following points are addressed, the manuscript can be published in Nature Communications.

Major points:

1. The effect of USP44 and TRIM25 on Ku80 expression is mainly investigated using exogenously expressing cells. The change in endogenous Ku80 expression in SUNE1 and HONE1 cells is only demonstrated by immunofluorescence. If western blot data showing the change at endogenous levels are available, the result could be confirmed.

Response:

Thanks for your valuable suggestions. Following the reviewer's suggestion, we conducted western blot assays in sgNC and sgUSP44 SUNE1 cells upon stable USP44 overexpression, as well as in sgNC and sgTRIM25 SUNE1 cells to see the changes in endogenous Ku80 expression upon CHX treatment. The results showed that knockout of USP44 inhibited the degradation of endogenous Ku80, which was consistent with that of exogenous Ku80 (previous data, moved as Supplementary Fig. 4d), indicating USP44 reduced the stability of Ku80 indeed in NPC cells (Page 6, paragraph 2; Page 7, paragraph 1; Page 31, paragraph 2; Fig. 3e). Furthermore, the same results were confirmed by knockout of TRIM25. The results showed that knockout of TRIM25 also inhibited the degradation of endogenous Ku80, which was consistent with that of exogenous Ku80 by knockdown of TRIM25 (previous data, moved as Supplementary Fig. 5e), indicating that TRIM25 promoted the degradation of Ku80 in NPC cells (Page 8, paragraph 2; Page 32, paragraph 1; Fig. 3k). Our results proved that USP44 and TRIM25 promoted the degradation of Ku80 both in the exogenous and endogenous levels in NPC cells.

Fig. 3 e, The effect of CHX treatment (left) and greyscale analysis of the results (right) in sgNC or sgUSP44 SUNE1 cells. The sgNC or sgUSP44 cells were constructed upon SUNE1 cells with stable USP44 overexpression.

k, The effect of CHX treatment (left) and greyscale analysis of the results (right) in sgNC or sgTRIM25 SUNE1 cells.

2. As shown in Suppl Fig 2, USP44 KD induces radiation resistance in SUNE1 and HONE1 cells. If Ku80 plays a direct role in radiation resistance, does silencing of Ku80 cancel radiation resistance in these cells? Alternatively, the same experiment can be done using USP44 KO cells.

Response:

Thanks for your valuable question and advice. Following the reviewer's suggestion, we conducted the clonogenic assays in SUNE1 and HONE1 cells transfected with the empty vector or USP44 plus the empty vector or Ku80 plasmids to see the effects of re-expression of Ku80 in USP44 overexpressing cells on radiation survival. The results showed that overexpression of USP44 in SUNE1 or HONE1 cells significantly inhibited cell survival after DNA damage caused by IR, which could be reversed by re-expression of Ku80 (Page 10, Paragraph 2; Supplementary Fig. 8e). Besides, we also conducted the clonogenic assays in USP44 WT or KO SUNE1 cells. The results showed that knockout of USP44 in SUNE1 cells promoted cell survival after DNA damage caused by IR (Page 5, Paragraph 2; Supplementary Fig. 2f). These results indicated that USP44's suppressive effects on NPC cell survival were dependent on Ku80 decrease.

Supplementary Fig. 8 e, Clonogenic assays (top), survival fraction curves (bottom) after exposure to indicated IR of SUNE1 and HONE1 cells transiently co-transfected with HA-USP44 or the empty vector plus FLAG-TRIM25 or the empty vector plasmids. Data are presented as the mean \pm SD, ** $P < 0.01$, *** $P < 0.001$ (two-tailed Student's t -test), $n = 3$ independent experiments performed in triplicate.

Supplementary Fig. 2 f, Clonogenic assays (f, left), survival fraction curves (f, right) after exposure to indicated IR of sgNC or sgUSP44 SUNE1 cells. The sgNC or sgUSP44 cells were constructed upon SUNE1 cells with stable USP44 overexpression. Data are presented as the mean \pm SD, *** $P < 0.001$ (two-tailed Student's t -test), $n = 3$ independent experiments performed in triplicate.

3. As shown in Fig. 7, both endogenous and exogenous expression levels of USP44 and TRIM25 are higher in irradiated cells than in non-irradiated cells even at 28 days, indicating that some mechanisms responding to radiation maintain their levels after irradiation. Is there any evidence suggesting that their levels are regulated by post-translational modifications associated with the DNA damage response? If not, this point should be discussed.

Response:

Thanks for your insightful comments and questions. Actually, we have noticed about this phenomenon that the USP44 and TRIM25 expression levels were higher in irradiated cells than in non-irradiated cells (Fig. 7d), which were confirmed by Western blot analysis (Supplementary Fig.

7d). Since USP44 increased the IR induced DNA double-strand break (DSB) (Supplementary Fig. 7d), we wonder if the damaged DNA would activate the anti-tumor immunity and further activate USP44 expression in NPC cells. So far, our results showed that knockout of USP44 enhanced the cGAS-STING signaling activated by IR (data not shown). Whether the activation of anti-tumor immunity is accounted for the increased expression of USP44 induced by IR is still not known. More workings are needed to explore the specific mechanisms in our further project. As for the increased expression of TRIM25 upon IR induction, this may be because of the change of USP44 expression, as we previously found that USP44 interacted with TRIM25 and then deubiquitinated and stabilized TRIM25 in NPC.

Supplementary Fig. 7 d, Western blot analysis of the Vector+sgNC, *USP44*+sgNC, and *USP44*+sg*TRIM25* grouped SUNE1 cells with IR treatment (6 Gy, 0.5 h) or not.

Minor points:

1. Figure 3c. Colocalization of Ku80 with HA-USP44 should be more clearly demonstrated.

Response:

Thanks for your valuable comments. It might cause misunderstandings for the reviewer because of the low-resolution images. Therefore, we repeated the immunofluorescence colocalization experiment in SUNE1 and HONE1 cells that transfected with HA-USP44 and improved the resolution of images. The new pictures were showed in fig. 3c, which more clearly demonstrated the colocalization of Ku80 with HA-USP44 in the nucleus (Fig. 3c).

Fig. 3 c, Immunofluorescence staining revealed the cellular location of exogenous HA-*USP44* (green) and endogenous Ku80 (red) at 0.5h after exposure to 6-Gy IR. The antibodies used were anti-HA antibody (H3663) and anti-Ku80 antibody (16389-1-AP). Scale bars, 10 μ m.

2. *Figure 3e & Figure 4b. Why are USP44 bands visible in USP44 KO cells?*

Response:

Thanks for your valuable questions. The USP44 KO cells were constructed upon *USP44* stably overexpression cells (because of the low expression of USP44 in NPC cells). We constructed three different sgRNAs targeting USP44, and these constructs were co-transfected into SUNE1 or HONE1 cells. Among twelve single colonies we generated, the single colony (in the figures) has the highest USP44 knockout efficiency.

Although our sgRNAs targeting USP44 did not fully block the expression of USP44, the results from clonogenic assay, western blotting, immunofluorescence assays showed that knockout of USP44 significantly reduced the radiosensitivity of NPC cells (Supplementary Fig. 2f), enhanced the stability of Ku80 (Fig. 3e and Supplementary Fig. 4d) and promoted the degradation of TRIM25 (Fig. 4b), indicating the sgRNAs targeting USP44 are functional.

In addition, there are some articles that show that sgRNA does not completely knock out the target protein¹⁻⁵. To address the reviewer's concerns, we have added the relevant description to our revised manuscript (Page 17, Paragraph 1).

Supplementary Fig. 2 f, Clonogenic assays (f, left), survival fraction curves (f, right) after exposure to indicated IR of sgNC or sgUSP44 SUNE1 cells. The sgNC or sgUSP44 cells were constructed upon SUNE1 cells with stable USP44 overexpression. Data are presented as the mean \pm SD, *** $P < 0.001$ (two-tailed Student's *t*-test), $n = 3$ independent experiments performed in triplicate.

Fig. 3 e, The effect of CHX treatment (left) and greyscale analysis of the results (right) in sgNC or sgUSP44 SUNE1 cells. The sgNC or sgUSP44 cells were constructed upon SUNE1 cells with stable USP44 overexpression.

Supplementary Fig. 4 d, The effect of CHX treatment (left) and greyscale analysis of the results (right) in sgNC or sgUSP44 SUNE1 cells transfected with FLAG-Ku80. The sgNC or sgUSP44 cells were constructed upon SUNE1 cells with stable USP44 overexpression.

Fig. 4 b, The effect of CHX treatment (left) and greyscale analysis of the results (right) in sgNC or sgUSP44 SUNE1 cells. The sgNC or sgUSP44 cells were constructed upon SUNE1 cells with stable USP44 overexpression.

References:

1. Liu, P.H. et al. An IRAK1-PIN1 signalling axis drives intrinsic tumour resistance to radiation therapy. *Nat. Cell Biol.* 21, 203-213 (2019).
2. Nakamura, K. et al. H4K20me0 recognition by BRCA1-BARD1 directs homologous recombination to sister chromatids. *Nat Cell Biol* 21, 311-318 (2019).
3. Liao, D. et al. Chromosomal translocation-derived aberrant Rab22a drives metastasis of osteosarcoma. *Nat Cell Biol* 22, 868-881 (2020).
4. Zhou, P. et al. MLL5 suppresses antiviral innate immune response by facilitating STUB1-mediated RIG-I degradation. *Nat Commun* 9, 1243 (2018).
5. Jia, M. et al. Redox homeostasis maintained by GPX4 facilitates STING activation. *Nat Immunol* 21, 727-735 (2020).

3. *Suppl Figure 2a. Is HA-USP44 correct in western blot? It may be USP44 to show the level in vector-transfected cells.*

Response:

Thanks for your valuable suggestion. Following the reviewer's suggestion, we conducted the western blot assays against anti-USP44 antibody in the vector and USP44 overexpression group in SUNE1 and HONE1 cells. The results were showed as following (Supplementary Fig. 2a).

Supplementary Fig. 2 a, Western blot analysis of SUNE1 and HONE1 cells stably overexpressing *USP44*.

4. *Reference 11. The page number is missing.*

Response:

Thanks for your kind reminding. It was our negligence that the page number of Reference 11 is missing. We have added the page number of Reference 11 in our revised manuscript (Page 24, paragraph 11).

REVIEWER COMMENTS

Reviewer #1 (Remarks to the Author):

In the revised manuscript, Chen et al. thoroughly addressed all my concerns. They have now provided additional experiments to address the problems raised by the previous version of this manuscript.

Although I am satisfied with how the authors addressed the concerns, there is one point that remains to be addressed in the current manuscript:

While the authors provided additional experiments to prove that USP44 deubiquitinates TRIM25 most efficiently when ubiquitinated on K439, their statements in lines 207 and 208 and 210 should change to accurately describe the presented data. Figure 4i shows that all mutants can be deubiquitinated by USP44, including K439R. Comparing the amount of K439-ubiquitin smears in cells expressing USP44 to the same signal in cells without USP44 expression clearly demonstrates that USP44 (reduces the signal) targets this mutant for deubiquitination as well (Fig. 4i compare lanes 3 and 7). Although the USP44 might have the highest activity toward K439 ubiquitinated, TRIM25 ubiquitinated on other lysines certainly does not appear resistant to USP44 deubiquitination.

This point needs to be reflected upon in the text before the publication of this work.

I accept the responses to all other concerns and am convinced that this work will be of great interest to Nature Communication readers.

Reviewer #2 (Remarks to the Author):

The authors have performed a substantial number of new experiments to address concerns that were raised by the reviewers. Most of my concerns have been adequately addressed, however I have a few additional/remaining comments.

First in the newly presented pH3S10 data (Suppl Figure 3c) I do not see striking differences between sgNC and sgUSP44 cells and quantification has not been performed to support differences. Additionally, pH3S10 has been used as a marker of cells in the M phase of the cell cycle (phosphorylation of H3 at S10 happens in mitosis) and in standard conditions IR would be expected to reduce staining of cells in the M phase (G2 arrest) to allow for repair of DNA prior to entering mitosis (PMID 11313470). In fact unrestrained progression of cells into mitosis with DNA damage present (defective G2 checkpoint) would be expected to have deleterious effects on cell survival. As such, perhaps rather than looking at pH3S10 intensity (unclear what this endpoint biologically represents) the authors should be investigating the percentage of pH3S10 positive cells to make conclusions regarding G2 vs M phase cells and the G2 cell cycle checkpoint.

Second, methodology regarding quantification of Ku80 fluorescent intensity (Figure 7b) has not been described (number of cells quantified per experiment, software used, etc).

Reviewer #3 (Remarks to the Author):

The manuscript is largely improved according to reviewers' comments. My major concerns regarding the significance of Ku80 in radiation resistance in this setting are addressed by additional experiments. Therefore, this manuscript can be considered for publication in Nature Communications.

Dear Editor and Reviewers,

Thank you very much for your insightful comments and suggestions, which have greatly helped us to improve our study. Enclosed is the revised version of manuscript (Ref.: NCOMMS-21-18892A) entitled, “**USP44 regulates irradiation-induced DNA double-strand break repair and suppresses tumorigenesis in nasopharyngeal carcinoma**”. We have revised the manuscript according to the reviewers’ comments and re-submitted the revised manuscript. We all appreciate your support and efforts on our manuscript and look forward to your further decision, and we sincerely hope to have the opportunity to publish this paper in *Nature Communications*.

The following is the point-by-point response to the reviewers’ comments and questions.

Reviewer #1 (Remarks to the Author):

In the revised manuscript, Chen et al. thoroughly addressed all my concerns. They have now provided additional experiments to address the problems raised by the previous version of this manuscript.

Although I am satisfied with how the authors addressed the concerns, there is one point that remains to be addressed in the current manuscript:

While the authors provided additional experiments to prove that USP44 deubiquitinates TRIM25 most efficiently when ubiquitinated on K439, their statements in lines 207 and 208 and 210 should change to accurately describe the presented data. Figure 4i shows that all mutants can be deubiquitinated by USP44, including K439R. Comparing the amount of K439-ubiquitin smears in cells expressing USP44 to the same signal in cells without USP44 expression clearly demonstrates that USP44 (reduces the signal) targets this mutant for deubiquitination as well (Fig. 4i compare lanes 3 and 7). Although the USP44 might have the highest activity toward K439 ubiquitinated, TRIM25 ubiquitinated on other lysines certainly does not appear resistant to USP44 deubiquitination.

This point needs to be reflected upon in the text before the publication of this work.

I accept the responses to all other concerns and am convinced that this work will be of great interest to Nature Communication readers.

Response:

Thanks for your valuable suggestion. Following the reviewer's suggestion, we redescribed the experimental results to make them more accurate. The results showed that without USP44 overexpression, the ubiquitination levels of all TRIM25 mutants (K283/284R, K439R or K509R) were weaker than that of the wild-type TRIM25 (WT), indicating that all the TRIM25 KR mutants can be ubiquitinated, including K439R. Comparing the amount of K439-ubiquitin smeared in cells with USP44 expressing to the same signal in cells without USP44 expressing clearly demonstrated that USP44 targeted this mutant for deubiquitylation as well. While TRIM25 ubiquitinated on other lysines certainly did not appear resistant to USP44 deubiquitylation, thus USP44 might have the highest activity toward TRIM25 K439 ubiquitination. We have corrected the description of Figure 4i in our revised manuscript (Page 9, paragraph 1).

Reviewer #2 (Remarks to the Author):

The authors have performed a substantial number of new experiments to address concerns that were raised by the reviewers. Most of my concerns have been adequately addressed, however I have a few additional/remaining comments.

1. First in the newly presented pH3S10 data (Suppl Figure 3c) I do not see striking differences between sgNC and sgUSP44 cells and quantification has not been performed to support differences. Additionally, pH3S10 has been used as a marker of cells in the M phase of the cell cycle (phosphorylation of H3 at S10 happens in mitosis) and in standard conditions IR would be expected to reduce staining of cells in the M phase (G2 arrest) to allow for repair of DNA prior to entering mitosis (PMID 11313470). In fact, unrestrained progression of cells into mitosis with DNA damage present (defective G2 checkpoint) would be expected to have deleterious effects on cell survival. As such, perhaps rather than looking at pH3S10 intensity (unclear what this endpoint biologically represents) the authors should be investigating the percentage of pH3S10 positive cells to make conclusions regarding G2 vs M phase cells and the G2 cell cycle checkpoint.

Response:

Thanks for your valuable suggestions. As the reviewer suggested, we analysed the percentage of H3S10_p positive cells rather than fluorescence intensity by immunofluorescence staining in sgNC or sgUSP44 SUNE1 cells treated with nocodazole before IR induction or not.

Although H3S10_p used to be regarded as a marker of cells in the M phase of the cell cycle (*PMID 11313470*), recent studies have found that it can also be detected in the interphase of the cell cycle¹⁻⁴. The microtubule poison nocodazole can arrest cells in G2/M phase when H3S10_p is highly abundant⁵⁻⁹. To circumvent the confusion caused by asynchronous cell populations in the interpretation of experimental results, we arrested cells in G2/M phase with nocodazole and investigated the effect of IR treatment and *USP44* knockout on G2/M cell cycle arrest.

Our results revealed that the percentage of H3S10_p positive cells was observably enhanced upon IR induction, which could lead to DNA damage and arrest cells in G2/M phase. Knockout of *USP44* decreased the percentage of H3S10_p positive cells and inhibited the IR-induced G2/M cell cycle arrest, which was consistent with the previous observations in the apoptosis experiments.

To address the reviewer's concerns, we have added these points in our revised manuscript (Page 6, paragraph 1; Supplementary Fig. 3c).

Supplementary Fig. 3 c, The percentage of H3S10_p positive cells was analyzed by immunofluorescence staining in sgNC or sgUSP44 SUNE1 cells treated with nocodazole (0.1 μg/ml) for 18 hours before IR treatment (6 Gy, 0.5 h) or not. Scale bar, 10μm. Data in a-c are presented as the mean ± SD, **P* < 0.05, ***P* < 0.01, ****P* < 0.001 (two-tailed Student's *t*-test), *n* = 3 independent experiments.

Reference:

1. Tjeertes, J.V., Miller, K.M. & Jackson, S.P. Screen for DNA-damage-responsive histone modifications identifies H3K9Ac and H3K56Ac in human cells. *EMBO J.* 28, 1878-1889 (2009).
2. Ruppert, J.G. et al. HP1α targets the chromosomal passenger complex for activation at heterochromatin before mitotic entry. *EMBO J.* 37 (2018).
3. Albig, C. et al. JASPer controls interphase histone H3S10 phosphorylation by chromosomal kinase JIL-1 in *Drosophila*. *Nat Commun* 10, 5343 (2019).
4. Baek, S.H. When signaling kinases meet histones and histone modifiers in the nucleus. *Mol. Cell* 42, 274-284 (2011).

5. Katada, H., Harumoto, T., Shigi, N. & Komiyama, M. Chemical and biological approaches to improve the efficiency of homologous recombination in human cells mediated by artificial restriction DNA cutter. *Nucleic Acids Res.* 40, e81 (2012).
6. Ogiwara, H. et al. Actin-related protein Arp4 functions in kinetochore assembly. *Nucleic Acids Res.* 35, 3109-3117 (2007).
7. Allison, S.J. & Milner, J. Loss of p53 has site-specific effects on histone H3 modification, including serine 10 phosphorylation important for maintenance of ploidy. *Cancer Res.* 63, 6674-6679 (2003).
8. Wang, H. et al. PLK1 targets CtIP to promote microhomology-mediated end joining. *Nucleic Acids Res.* 46, 10724-10739 (2018).
9. Matsui, Y., Nakayama, Y., Okamoto, M., Fukumoto, Y. & Yamaguchi, N. Enrichment of cell populations in metaphase, anaphase, and telophase by synchronization using nocodazole and blebbistatin: a novel method suitable for examining dynamic changes in proteins during mitotic progression. *Eur. J. Cell Biol.* 91, 413-419 (2012).

2. *Second, methodology regarding quantification of Ku80 fluorescent intensity (Figure 7b) has not been described (number of cells quantified per experiment, software used, etc).*

Response:

Thanks for your valuable comments. It is our negligence not to describe the methodology regarding quantification of Ku80 fluorescent intensity in Supplementary Fig. 7b. The mean intensity of Ku80 fluorescence ($n = 20$ cells per group) in Fig. 4j was quantified using ImageJ software. Data in Supplementary Fig. 7b are presented as the mean \pm SD, $***P < 0.001$ (two-tailed Student's t -test) of 3 independent experiments. To address the reviewer's concerns, we have added these points in our revised Supplementary information (Supplementary Fig. 7b). The statistical methods are described at the end of the legend in Supplementary Fig. 7, just like other Figure legend parts of our manuscript.

Reviewer #3: (Remarks to the Author):

The manuscript is largely improved according to reviewers' comments. My major concerns regarding the significance of Ku80 in radiation resistance in this setting are addressed by additional experiments. Therefore, this manuscript can be considered for publication in *Nature Communications*.

Response:

Thank you for being satisfied with our revisions. We all truly enjoyed the stimulating

interactions with you.

REVIEWERS' COMMENTS

Reviewer #2 (Remarks to the Author):

The authors have been responsive to the reviewers and I believe this paper is now suitable for publication in Nature Communications.

REVIEWERS' COMMENTS

Reviewer #2 (Remarks to the Author):

The authors have been responsive to the reviewers and I believe this paper is now suitable for publication in Nature Communications.